# An analytic theory for the degree of Arctic Amplification

Wenyu Zhou [1] ✉, L. Ruby Leung [1], Shang-Ping Xie [2] & Jian Lu [1]

Arctic Amplification (AA), the amplified surface warming in the Arctic relative to the globe, is a salient feature of climate change. While the basic physical picture of AA has been depicted, how its degree is determined has not been clearly understood. Here, by deciphering atmospheric heat transport (AHT), we build a two-box energy-balance model of AA and derive that the degree of AA is a simple nonlinear function of the Arctic and global feedbacks, the meridional heterogeneity in radiative forcing, and the partial sensitivities of AHT to global mean and meridional gradient of warming. The formula captures the varying AA in climate models and attributes the spread to models' feedback parameters and AHT physics. The formula clearly illustrates how essential physics mutually determine the degree of AA and limits its range within 1.5-3.5. Our results articulate AHT as both forcing and feedback to AA, highlight its fundamental role in forming a baseline AA that exists even with uniform feedbacks, and underscore its partial sensitivities instead of its total change as key parameters of AA. The lapse-rate feedback has been widely recognized as a major contributor to AA but its effect is fully offset by the water-vapor feedback.

Robustly seen in paleo proxy records[1,2], historical observations[3–5], and model simulations[6–9], the temperature response to climate change is amplified in the Arctic relative to the rest of Earth. This so-called Arctic Amplification (AA) not only affects the Arctic cryosphere and ecosystem but also influences global circulation and climate by modulating the meridional temperature gradient[10–16]. However, the degree of AA, commonly defined as the ratio in surface temperature changes between the Arctic and global mean, varies widely by a factor of two among climate models[17]. This large uncertainty is not unexpected, considering the complex processes and feedbacks contributing to AA[17–19]. Locally, surface warming in the Arctic is amplified by the positive albedo feedback due to sea ice loss[20,21], the positive lapse-rate feedback due to surface-trapped warming[18,22], and the less negative Planck feedback due to colder climate[18]. Remotely, the increase in meridional atmospheric heat transport (AHT), implied by enhanced meridional moisture gradient from global-scale warming, must be constrained by amplified Arctic warming so that it does not exceed the demand from the meridional gradient in radiative forcing[23–28].

Furthermore, the above effects are not independently additive but can interact with each other[29–31].

The relation between AA and its contributing factors has been studied through diagnostic and modelling methods. For example, the contributions of individual factors to Arctic and global warming have been diagnosed based on energy budget equations[32–36]. The dependences of AA on regional feedbacks, ocean heat flux, and radiative forcing have been investigated using both the diffusive energy balance model (EBM)[37–40] and comprehensive climate models[41]. For AA in a diffusive EBM with uniform feedbacks, analytic estimates have been proposed[39]. Despite these achievements, our understanding of how the degree of AA is determined remains limited as several outstanding questions remain: What are the essential physics that determine the degree of AA? What is the functional relationship between the degree of AA and the key physical parameters? How to understand the range of AA (i.e., why it is 1.5–3.5 instead of 10)? What causes the variation in the degree of AA among models?

[1]Atmospheric, Climate and Earth Sciences Division, Pacific Northwest National Laboratory, Richland, WA, USA. [2]Scripps Institution of Oceanography, University of California San Diego, La Jolla, CA, USA. ✉e-mail: wenyu.zhou@pnnl.gov

Here, we answer the above questions by establishing an analytic theory for the degree of AA. Specifically, we build a two-box energy-balance model of AA and reveal that the degree of AA is a simple nonlinear function of five key physical parameters. The theoretical function conveys a concise picture of how the degree of AA emerges from essential physics and explains the varying degree of AA in individual climate models. Furthermore, it clearly interprets the intricate role of AHT in AA and reveals the close compensation between the effects of the lapse-rate and water-vapor feedbacks on AA.

## Results

### The degree of AA as a simple nonlinear function of key physical parameters

We start with the energy balance equations for the Arctic and global mean under climate change,

$$Arctic: F^A + \Delta O^A + \Delta AHT + \sum_i \lambda_i^A \Delta T^A = 0, \tag{1}$$

$$Global: F^G + \Delta O^G + \sum_i \lambda_i^G \Delta T^G = 0, \tag{2}$$

where the superscripts $A$ and $G$ denote the Arctic and global mean respectively, $F$ is the effective radiative forcing, $\Delta$ denotes the response to climate change, $T$ is the surface temperature, $\lambda_i$ represents individual feedback parameters (defined respectively with respect to the Arctic and global mean surface warming) from Planck, lapse rate, water vapor, albedo and clouds, $\Delta O$ refers to the change in oceanic heat flux, and $\Delta AHT$ represents the change in AHT into the Arctic.

The widely-used diagnostic method[32–36] utilizes Eqs. (1, 2) to partition the total warming into contributions of individual factors as

$$\Delta T^A = \frac{F^A}{-\lambda_P^G} + \frac{\Delta O^A}{-\lambda_P^G} + \frac{\Delta AHT}{-\lambda_P^G} + \frac{(\lambda_{alb}^A + \lambda_{LR}^A + \lambda_{WV}^A + \lambda_{cld}^A + \lambda_P'^A)\Delta T^A}{-\lambda_P^G}, \tag{3}$$

$$\Delta T^G = \frac{F^G}{-\lambda_P^G} + \frac{\Delta O^G}{-\lambda_P^G} + \frac{(\lambda_{alb}^G + \lambda_{LR}^G + \lambda_{WV}^G + \lambda_{cld}^G)\Delta T^G}{-\lambda_P^G}, \tag{4}$$

where $\lambda_P^G$ is the global mean Planck feedback and $\lambda_P'^A = \lambda_P^A - \lambda_P^G$. The feedback $\lambda_i$ is diagnosed as contributing to the total Arctic warming ($\Delta T^A$) by $-\frac{\lambda_i^A \Delta T^A}{\lambda_P^G}$ and to the total global warming ($\Delta T^G$) by $-\frac{\lambda_i^G \Delta T^G}{\lambda_P^G}$. Such partition is helpful for understanding the relative importance of individual factors. However, with $\Delta T^A$ and $\Delta T^G$ used as input, it is silent on why the degree of AA varies among models, and more generally, how the degree of AA varies with these physical factors. Furthermore, the attributed warming may not reflect the real role of a specific factor. For example, if a feedback has the same strength between the Arctic and global mean ($\lambda_i^A = \lambda_i^G$), it should not physically contribute to AA but would be diagnosed as contributing more warming to the Arctic relative to the globe given $\Delta T^A > \Delta T^G$. A theory, beyond the current diagnostic framework, is needed to better understand how the degree of AA is determined.

Here, we build a two-box energy-balance model of AA by deciphering the change in AHT and upon that derive an analytic formula between AA and its contributing factors (Fig. 1). The change in AHT, $\Delta AHT$, is a unique factor involved in AA. It is neither a pure forcing (e.g., $F$) nor a pure feedback (e.g., $\lambda_i$) to AA. Instead, it consists of two parts− a forcing-like part as global-scale warming enhances meridional moisture gradient ($dq^*/dT$ is higher in warmer tropics) and increases AHT to amplify Arctic warming, and a negative feedback part as AA weakens meridional temperature gradient and reduces AHT. This motivates us to formulate $\Delta AHT$ as a function of global mean warming

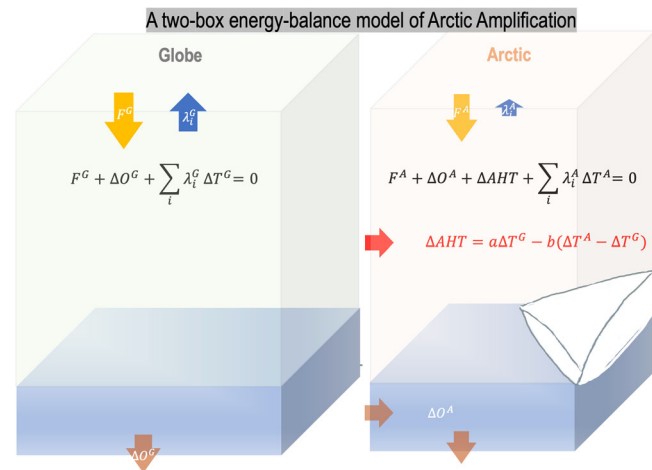

**Fig. 1 | A two-box energy-balance model of Arctic Amplification (AA).** The two boxes represent the global mean and the Arctic mean, respectively. The energy balance equations involve forcing ($F$), feedbacks ($\sum_i \lambda_i \Delta T$), the change in ocean heat convergence and uptake ($\Delta O$), and the change in atmospheric heat transport into the Arctic ($\Delta AHT$). $\Delta AHT$ can be formulated as a function of global mean warming ($\Delta T^G$) and enhanced Arctic warming relative to the global mean ($\Delta T^A - \Delta T^G$).

$\Delta T^G$ and enhanced Arctic warming $\Delta T^A - \Delta T^G$,

$$\Delta AHT \cong a\Delta T^G - b\left(\Delta T^A - \Delta T^G\right), \tag{5}$$

and a reformulation of Eq. (5) gives

$$\frac{\Delta AHT}{\Delta T^G} \cong a - b(AA - 1), \tag{6}$$

The above formula of $\Delta AHT$ is not only supported by the significant correlation between $\frac{\Delta AHT}{\Delta T^G}$ and $AA - 1$ across models ($r = -0.70$; Fig. 2a) but also derivable from the theoretical diffusive formulation of AHT (Methods). The intermodel regression between $\frac{\Delta AHT}{\Delta T^G}$ and $AA - 1$ estimates $\hat{a} \cong 2.1\,W\,m^{-2}\,K^{-1}$ and $\hat{b} \cong 1.7\,W\,m^{-2}\,K^{-1}$ (Fig. 2a). The diffusive theory derives $a$ and $b$ as functions of basic parameters of the climate systems and suggests consistent values (Methods). The hat symbol here denotes a general, constant estimate for the model ensemble instead of individual models. With constant estimates of $\hat{a}$ and $\hat{b}$, Eq. (5) reasonably captures $\Delta AHT$ projected by models ($r = 0.71$), explaining ~50% of the variance in $\Delta AHT$ (Fig. S1). The unexplained variance reflects the variations in the parameters $a \equiv \frac{\partial AHT}{\partial T^G}$ and $b \equiv -\frac{\partial AHT}{\partial (T^A - T^G)}$ among models, that is, the model-dependent sensitivities of AHT to global uniform warming and enhanced Arctic warming. We further estimate $a$ and $b$ for individual models according to the deviation of $\frac{\Delta AHT}{\Delta T^G}$ from the regression line, i.e., $\frac{\Delta \hat{AHT}}{\Delta T^G} = \hat{a} - \hat{b}(AA - 1)$, so that Eq. (5) reproduces $\Delta AHT$ in individual models (Methods).

To facilitate derivation, we have also written the change in ocean heat flux, $\Delta O$, as a feedback to the temperature change, similar to ref. 42,

$$\Delta O = \lambda_O \Delta T, \tag{7}$$

where $\lambda_O$ is diagnosed as $\Delta O/\Delta T$ for the globe and the Arctic. While this is mainly a mathematical treatment, it can be physically justified as $\Delta O$ is approximately proportional to $\Delta T$ over the transient response to climate change (Fig. S2).

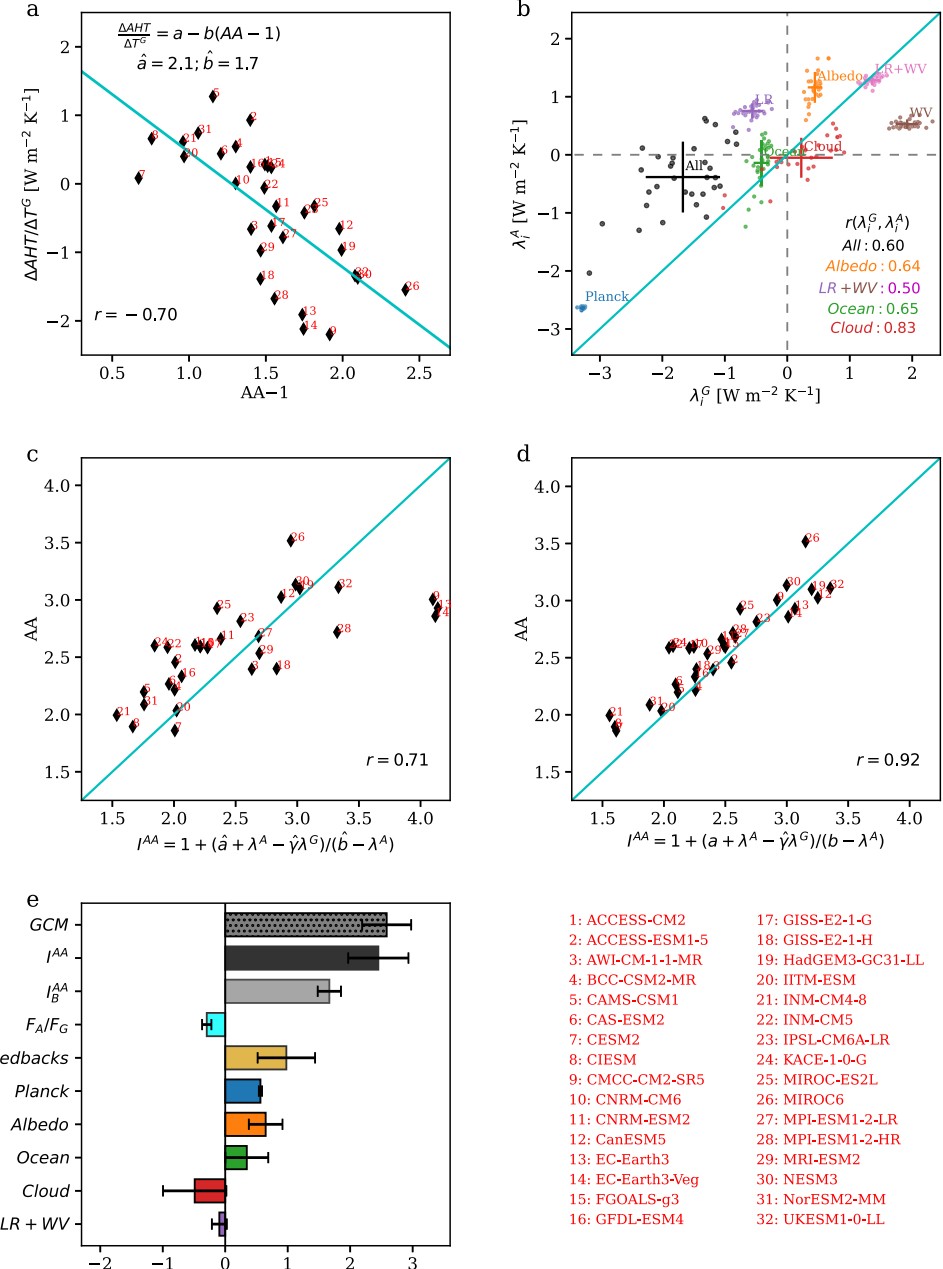

**Fig. 2 | Development of an analytic formula for the degree of Arctic Amplification (AA). a** Scatterplot between AA−1 and the change in atmospheric heat transport into the Arctic normalized by the global mean warming ($\Delta AHT / \Delta T^G$) across models. **b** The global and Arctic feedback parameters (defined with respect to their respective surface warming) diagnosed using the radiative-kernel method. The crosses indicate the uncertainty range (±s.d.) across models and the dots indicate values in individual modes. The correlations between the global and Arctic feedbacks are denoted for the total and individual feedbacks. **c.** Scatterplot between the model-projected AA and the theoretical estimate Eq. (8) using constant $\hat{a}$ and $\hat{b}$. **d** Scatterplot between the model-projected AA and the theoretical estimate Eq. (8) using model-dependent $a$ and $b$. **e** The mean and uncertainty (±s.d.) of the model-projected AA, the theoretical estimate $I^{AA} \equiv 1 + \frac{a + \lambda^A - \hat{\gamma}\lambda^G}{b - \lambda^A}$ Eq. (8), the baseline AA $I_B^{AA} \equiv 1 + \frac{a}{b - \lambda^G}$ Eq. (9), and the effects from differential forcing ($\gamma<1$), differential total feedback ($\lambda^A > \lambda^G$) and differential individual feedbacks ($\lambda_i^A$ vs. $\lambda_i^G$) between the global and the Arctic mean. Source data are provided as a Source Data file.

Equations (1, 2) and (5–7) form a two-box energy-balance model of AA, in which the Arctic and the globe experience their respective forcing and feedbacks and interact through AHT (see Fig. 1 for a schematic). This two-box model yields a theoretical solution for the degree of AA as,

$$AA \equiv \frac{\Delta T^A}{\Delta T^G} = 1 + \frac{a + \lambda^A - \gamma\lambda^G}{b - \lambda^A} \equiv I^{AA}, \qquad (8)$$

where $\gamma \equiv \frac{F^A}{F^G}$ is the ratio in radiative forcing between the Arctic and the globe, $\lambda^A \equiv \sum \lambda_i^A$ and $\lambda^G \equiv \sum \lambda_i^G$ are the sums of feedbacks (Planck, albedo, lapse rate, water vapor, cloud, and ocean) for the Arctic and the globe respectively, $a \equiv \frac{\partial AHT}{\partial T^G}$ measure the increasing rate of AHT with global uniform warming and $b \equiv -\frac{\partial AHT}{\partial(T^A - T^G)}$ measures the decreasing rate of AHT with enhanced Arctic warming. Physically, the numerator $a + \lambda^A - \gamma\lambda^G$ measures the anomalous energy input into the

Arctic relative to the globe because of the increasing AHT with global uniform warming and its less negative feedback, while the denominator $b - \lambda^A$ measures the energy damping efficiency of the Arctic, reflected by the decreasing AHT with enhanced Arctic warming minus the Arctic feedback.

We proceed to validate the theory by applying it to capture the degree of AA in individual climate models (Methods). The responses to climate change ($\Delta$) are computed from the differences between the 1980−1995 period in the historical simulations and the 2085−2100 period under the Shared Socioeconomic Pathways SSP2-4.5 in 32 models from the Coupled Model Intercomparison Project Phase 6 (CMIP6)[43]. The feedback parameters, $\lambda_i$, are diagnosed using the radiative-kernel method[44,45] for both the globe and the Arctic. As shown in Fig. 2b, the Planck, albedo, lapse-rate feedbacks are less negative or more positive in the Arctic than in the globe (above the 1:1 line) and would contribute positively to AA; in contrast, the cloud and water-vapor feedbacks contribute negatively to AA. It is difficult to accurately estimate the parameter $\gamma$ for individual models, so we adopt a constant value of $\hat{\gamma} = 0.6$ based on the spatial pattern of the 2xCO2 radiative forcing[46]. The omission of a model-dependent $\gamma$ may be justified as radiative forcing accounts for only a small portion of uncertainty in the absolute Arctic warming[18,36]. Using diagnosed model-dependent $\lambda^A$ and $\lambda^G$ but constant $\hat{a}$, $\hat{b}$ and $\hat{\gamma}$, the theoretical formula reasonably captures the degree of AA in individual models, with $I^{AA} = 1 + \frac{\hat{a} + \lambda^A - \hat{\gamma}\lambda^G}{\hat{b} - \lambda^A}$ correlated with the model-projected AA at $r = 0.71$ across models (Fig. 2c). The accuracy is further improved by using the model-dependent $a$ and $b$. $I^{AA} = 1 + \frac{a + \lambda^A - \hat{\gamma}\lambda^G}{b - \lambda^A}$ is correlated with the model-projected AA at $r = 0.92$ across models (Fig. 2d).

## A concise picture of how the degree of AA emerges from essential physics

The theoretical formula, $I^{AA} \equiv 1 + \frac{a + \lambda^A - \gamma\lambda^G}{b - \lambda^A}$, conveys a concise picture of how the degree of AA is mutually determined by essential physics. First, a baseline AA exists with even spatially uniform radiative forcing ($\gamma = 1$) and feedbacks ($\lambda^A = \lambda^G$). This baseline AA,

$$I_B^{AA} = 1 + \frac{a}{b - \lambda^G}, \tag{9}$$

reflects the fundamental role of AHT and is set by the ratio between the increasing rate of AHT with global uniform warming ($a$) and the decreasing rate of AHT with enhanced Arctic warming minus global climate feedback ($b - \lambda^G$). Its degree is estimated to be $1.67 \pm 0.19$ ($\pm$s.d.) (grey bars in Fig. 2e) based on the values of $a$, $b$, and $\lambda^G$. Then, the differential radiative forcing and feedbacks between globe and Arctic influences AA by

$$\begin{aligned} FF \equiv I^{AA} - I_B^{AA} &= \frac{a + \lambda^A - \gamma\lambda^G}{b - \lambda^A} - \frac{a}{b - \lambda^G} \\ &= \frac{a + \lambda^A - \lambda^G + (1 - \gamma)\lambda^G}{b - \lambda^G - (\lambda^A - \lambda^G)} - \frac{a}{b - \lambda^G}. \end{aligned} \tag{10}$$

Specifically, the weaker radiative forcing in the Arctic relative to the globe reduces AA ($\gamma \equiv \frac{F^A}{F^G} < 1$ with $\lambda^G < 0$; cyan bar in Fig. 2e) but the less negative (or even positive) climate feedback in the Arctic amplifies AA ($\lambda^A - \lambda^G > 0$; golden bar in Fig. 2e). The effect of differential feedbacks dominates and the degree of AA is amplified from the baseline AA to $2.51 \pm 0.49$. We further quantify the contribution of individual feedback to the degree of AA by overriding its Arctic value with its global value in the formula (Methods). The Planck, albedo and ocean feedbacks increase the degree of AA respectively by $0.57 \pm 0.03$,

$0.65 \pm 0.28$ and $0.35 \pm 0.34$, while the cloud feedback decreases AA by $0.49 \pm 0.51$.

The lapse-rate feedback, being negative in low latitudes with amplified upper-level warming but positive in the Arctic with surface-trapped warming, has been widely recognized as a major contributor to AA[18]. Furthermore, as lapse rate in the Arctic is affected by radiative forcing and AHT, the lapse-rate feedback presents dependency on other factors involved in AA[29–31]. Understanding the lapse-rate feedback and its intricacy seems to be a leading-order problem for understanding AA. On the other hand, it has been long recognized in the climate sensitivity research community that the lapse-rate feedback is compensated by the water-vapor feedback[47]. Here, we show that the contribution of the lapse-rate feedback to AA is fully, if not overly, compensated by that of the water-vapor feedback (Fig. 2e). As the surface-trapped warming simultaneously reduces the increase in the tropospheric water vapor (Fig. 3a, b), the more positive lapse-rate feedback in the Arctic is associated with a less positive water-vapor feedback, so the sum of these two feedbacks is nearly identical between the globe and the Arctic (Fig. 2b; Fig. 3c). Furthermore, models that simulate a larger tropospheric warming also present a larger tropospheric moistening (solid versus dashed line in Fig. 3a, b), so the lapse-rate and water-vapor feedbacks are negatively correlated across models for both the globe ($r = -0.74$) and the Arctic ($r = -0.88$), leading to small intermodel spread in their sums (Figs. 2b, 3c). Eventually, the sum of the lapse-rate and water-vapor feedbacks is $1.36 \pm 0.11 \, \text{W m}^{-2} \, \text{K}^{-1}$ for the globe and $1.28 \pm 0.07 \, \text{W m}^{-2} \, \text{K}^{-1}$ for the Arctic. Thus, the combined feedback contributes little to AA and accounts for little intermodel uncertainty in AA compared to other feedbacks (Fig. 2e).

The formula $I^{AA} = 1 + \frac{a + \lambda^A - \gamma\lambda^G}{b - \lambda^A}$ articulates the intricate role of AHT in AA. In literature, the effect of AHT on AA is often diagnosed from $\Delta$AHT (see Eq. (3)). $\Delta$AHT, however, consists of a negative feedback part that depends on the degree of AA itself. The formula here highlights the partial sensitivities of AHT to global uniform warming ($a \equiv \frac{\partial AHT}{\partial T^G}$) and meridional warming gradient ($b \equiv -\frac{\partial AHT}{\partial (T^A - T^G)}$), instead of $\Delta$AHT itself, as the fundamental parameters for understanding the role of AHT. Even without differential feedbacks, the physics of AHT leads to the baseline AA, $I_B^{AA} = 1 + \frac{a}{b - \lambda^G} \cong 1.67$, which forms a major part of the total AA. The skill of $I^{AA}$ in capturing AA is improved from $r = 0.71$ to $r = 0.92$ by using the model-dependent $a$ and $b$, highlighting the important role of AHT in determining the degree of AA in individual models. The effect of AHT in individual models can be further understood from the following approximated formula (see derivation in Methods),

$$I^{AA} = 1 + \frac{a + \lambda^A - \gamma\lambda^G}{b - \lambda^A} \cong 1 + \frac{\hat{a} + d + \lambda^A - \gamma\lambda^G}{\hat{b} - \lambda^A}, \tag{11}$$

where $d = \frac{\Delta AHT}{\Delta T^G} - \widehat{\frac{\Delta AHT}{\Delta T^G}}$ is the difference between the model-simulated $\frac{\Delta AHT}{\Delta T^G}$ and the prior-estimated $\widehat{\frac{\Delta AHT}{\Delta T^G}} = \hat{a} - \hat{b}(AA - 1)$ according to $\hat{a}$ and $\hat{b}$ (i.e., the regression line in Fig. 2a). The effect of AHT is represented by $\hat{a}$, $\hat{b}$, and $d$. A more positive $d = \frac{\Delta AHT}{\Delta T^G} - \widehat{\frac{\Delta AHT}{\Delta T^G}}$ implies a higher $a$ and/or a lower $b$ that would favor a higher AA. This further explains why directly using $\Delta$AHT to interpret the effect of AHT may be misleading. For example, as shown in Fig. 2a, M12 (CanESM5) projects a negative $\Delta$AHT but has a positive $d$ that would favor a larger AA.

The formula $I^{AA} \equiv 1 + \frac{a + \lambda^A - \gamma\lambda^G}{b - \lambda^A}$ provides a physical explanation for the likely range of AA, specifically, why AA sits roughly between 1.5 and 3.5 in climate models. The partial sensitivities of AHT to global uniform warming ($a$) and enhanced Arctic warming ($b$) are rooted in basic parameters of the climate system (Methods) and estimated to be $a = 2.1 \pm 0.3 \, \text{W m}^{-2} \, \text{K}^{-1}$ and $b = 1.7 \pm 0.3 \, \text{W m}^{-2} \, \text{K}^{-1}$ in models. The

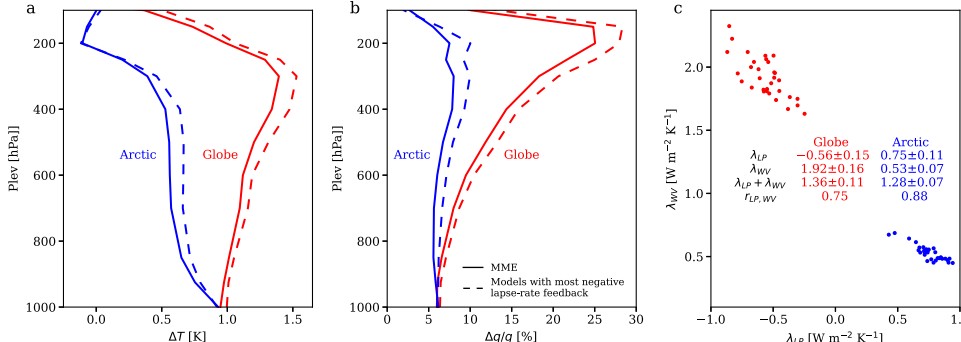

**Fig. 3 | Compensation between effects of the lapse-rate and water-vapor feedbacks on Arctic Amplification (AA). a** The vertical profile of the change in air temperature for the globe (red) and the Arctic (blue). **b** The vertical profile of the fractional change in specific humidity for the globe (red) and the Arctic (blue). In (**a** and **b**), the solid lines show the mean of the model ensemble (MME) and the dashed lines show the mean of five models with more negative lapse-rate feedback. **c** Intermodel scatterplot between the lapse-rate feedback ($\lambda_{LP}$) and the water-vapor feedback ($\lambda_{WV}$) in the globe (red) and the Arctic (blue). The mean and s.d. of $\lambda_{LP}$, $\lambda_{WV}$, $\lambda_{LP} + \lambda_{WV}$, and the intermodel correlation between $\lambda_{LP}$ and $\lambda_{WV}$ are listed for the globe and the Arctic. Source data are provided as a Source Data file.

global and Arctic climate feedbacks, as diagnosed from the radiative-kernel method, are $\lambda^G = -1.7 \pm 0.6$ W m$^{-2}$ K$^{-1}$ and $\lambda^A = -0.4 \pm 0.6$ W m$^{-2}$ K$^{-1}$. Furthermore, the diffusive theory derives that $a/b$ should be around one given the climatological distribution of specific humidity (Eq. (27); Methods), and the positive correlation between $\lambda^A$ and $\lambda^G$ (Fig. 2b) constrains $\lambda^A - \gamma\lambda^G$ to be $0.6 \pm 0.5$ W m$^{-2}$ K$^{-1}$. Applying these ranges in $I^{AA}$ suggests that it is very hard for the degree of AA to be below 1.5 or above 3.5.

## Understand the variation and outliers of AA among climate models

As the theory well captures the degree of AA in individual models (Fig. 2d), we further apply it to understand the causes of the variation and outliers of AA. In Fig. 4a, the theory-predicted AA, $I^{AA} \equiv 1 + \frac{a + \lambda^A - \hat{\gamma}\lambda^G}{b - \lambda^A}$, is decomposed into the sum of the baseline AA, $I_B^{AA}$ Eq. (9) and the effect of differential forcing and feedback between Arctic and globe, $FF$ Eq. (10). Models are ranked according to $I^{AA}$ and the magnitude of $FF$ is indicated by the cool-to-warm color. The outlier models, three with the lowest $I^{AA}$ (M7, M8, M21) and three with the highest $I^{AA}$ (M12, M19, M32), are denoted. The variation in $I^{AA}$ are contributed by both $I_B^{AA}$ (1.3–2.1) and $FF$ (0–1.8), but the effect of $FF$ is more dominating. To understand why the degree of AA varies among models, we proceed to understand the variations in $I_B^{AA}$ and $FF$.

According to Eq. (9), $I_B^{AA}$ is larger if $a \equiv \frac{\partial AHT}{\partial T^G}$ is larger, $b \equiv -\frac{\partial AHT}{\partial (T^A - T^G)}$ is smaller, and $\lambda^G$ is weaker (less negative). A model that simulates a higher $\frac{\Delta AHT}{\Delta T^G} = a - b(AA - 1)$ than the prior-estimated $\widehat{\frac{\Delta AHT}{\Delta T^G}} = \hat{a} - \hat{b}(AA - 1)$, should have a larger $a$ and/or a smaller $b$. Thus, $I_B^{AA}$ would be larger if $d = \frac{\Delta AHT}{\Delta T^G} - \widehat{\frac{\Delta AHT}{\Delta T^G}}$ is higher and $\lambda^G$ is weaker. Indeed, as shown in Fig. 4b, the intermodel variation of $I_B^{AA}$ (indicated by cool-to-warm color of the diamond symbols) is consistently explained by its increases with $d$ and $\lambda^G$. For the outlier models, M7 presents a small $I_B^{AA}$ due to its abnormally small $d$; M21 presents a small $I_B^{AA}$ due to its excessively strong $\lambda^G$ (<−3 W m$^{-2}$); on the other hand, M12,19,32 present a large $I_B^{AA}$ due to their weak $\lambda^G$ (>−1 W m$^{-2}$).

According to Eq. (10), $FF$ would be larger when $\lambda^G$ is weaker (less negative) but $\lambda^A - \lambda^G$ is higher (more positive). We relate $FF$ to $\lambda^G$ and $\lambda^A - \lambda^G$ instead to $\lambda^A$ and $\lambda^G$, as there is a significant correlation between $\lambda^A$ and $\lambda^G$ ($r = 0.6$; black dots in Fig. 2b). As shown in Fig. 4c, the intermodel variation of $FF$ (indicated by cool-to-warm color of the diamond symbols) is consistently explained by its increases with $\lambda^G$ and

$\lambda^A - \lambda^G$. For the outlier models, M7 and M8 present a very small $FF$ as their $\lambda^A - \lambda^G$ is abnormally low; M21 presents a very small $FF$ despite normal $\lambda^A - \lambda^G$ as its $\lambda^G$ is excessively strong; on the other hand, M12, 19, 32 present a strong $FF$ as their $\lambda^G$ are abnormally weak.

The variations in $\lambda^A - \lambda^G$ and $\lambda^G$ are further attributed to the effects of individual feedbacks ($\lambda_i^A$ and $\lambda_i^G$). Based on their baseline AA, $FF$, and $d$, $\lambda_i^A, \lambda_i^G$ (Fig. 4d), we conclude the causes of the abnormal AA in the six outlier models as follows.

- M7 (CESM2): Small baseline AA due to abnormally low $d$; Small $FF$ due to low $\lambda^A - \lambda^G$ which is further attributed to the effects of $\lambda_{Cld}$ and $\lambda_{Ocn}$.
- M8 (CIESM): Small $FF$ due to abnormally low $\lambda^A - \lambda^G$, which is further attributed to the effects of $\lambda_{Alb}$ and $\lambda_{Ocn}$.
- M21 (INM-CM4-8): Small baseline AA and $FF$ due to excessively strong $\lambda^G$, which is further attributed to the effect of $\lambda_{Cld}$.
- M12 (CanESM5), M19 (HadGEM3-GC31-LL), M32 (UKESM1-0-LL): Large baseline AA and $FF$ due to abnormally weak $\lambda^G$, which is further attributed to the effect of $\lambda_{Cld}$.

Thus, the intermodel spread in the degree of AA can be understood from three key parameters, the dynamics of AHT ($d$), the differential feedbacks between Arctic and globe ($\lambda^A - \lambda^G$), and the global mean feedback ($\lambda^G$). Given the formula $I^{AA} \equiv 1 + \frac{a + \lambda^A - \gamma\lambda^G}{b - \lambda^A}$, one might expect AA to increase with a stronger (more negative) $\lambda^G$. However, as $\lambda^A$ and $\lambda^G$ are correlated, a stronger $\lambda^G$ implies a stronger $\lambda^A$ and reduces $I^{AA}$. By influencing $\lambda^G$, global mean cloud feedback strongly affects the degree of AA, as seen in the outlier models M21 and M12,19 and 32.

## Application to explaining previous numerical results of AA

The formula, $I^{AA} \equiv 1 + \frac{a + \lambda^A - \gamma\lambda^G}{b - \lambda^A}$, can analytically explain the degree of AA in previous numerical studies that manipulate feedbacks and radiative forcing. Reference 39 imposed uniform forcing and feedbacks in a diffusive EBM. The resultant AA is ~1.6, which is consistent with the baseline AA here. Reference 41 investigated AA under regionally confined radiative forcing in a comprehensive climate model. They found a strong AA with polar forcing (poleward of 60°) but a very weak AA with tropical or midlatitudinal forcing. These results can be analytically explained by our formula. Specifically, $I^{AA} \cong \frac{a + b - \gamma\lambda^G}{b - \lambda^A}$ predicts $I^{AA} \cong 6$ for polar forcing with $\gamma \cong 4$ and $I^{AA} \cong 1.7$ for tropical or midlatitudinal forcing with $\gamma = 0$. Reference 40 conducted feedback-removing experiments in a diffusive EBM and showed that

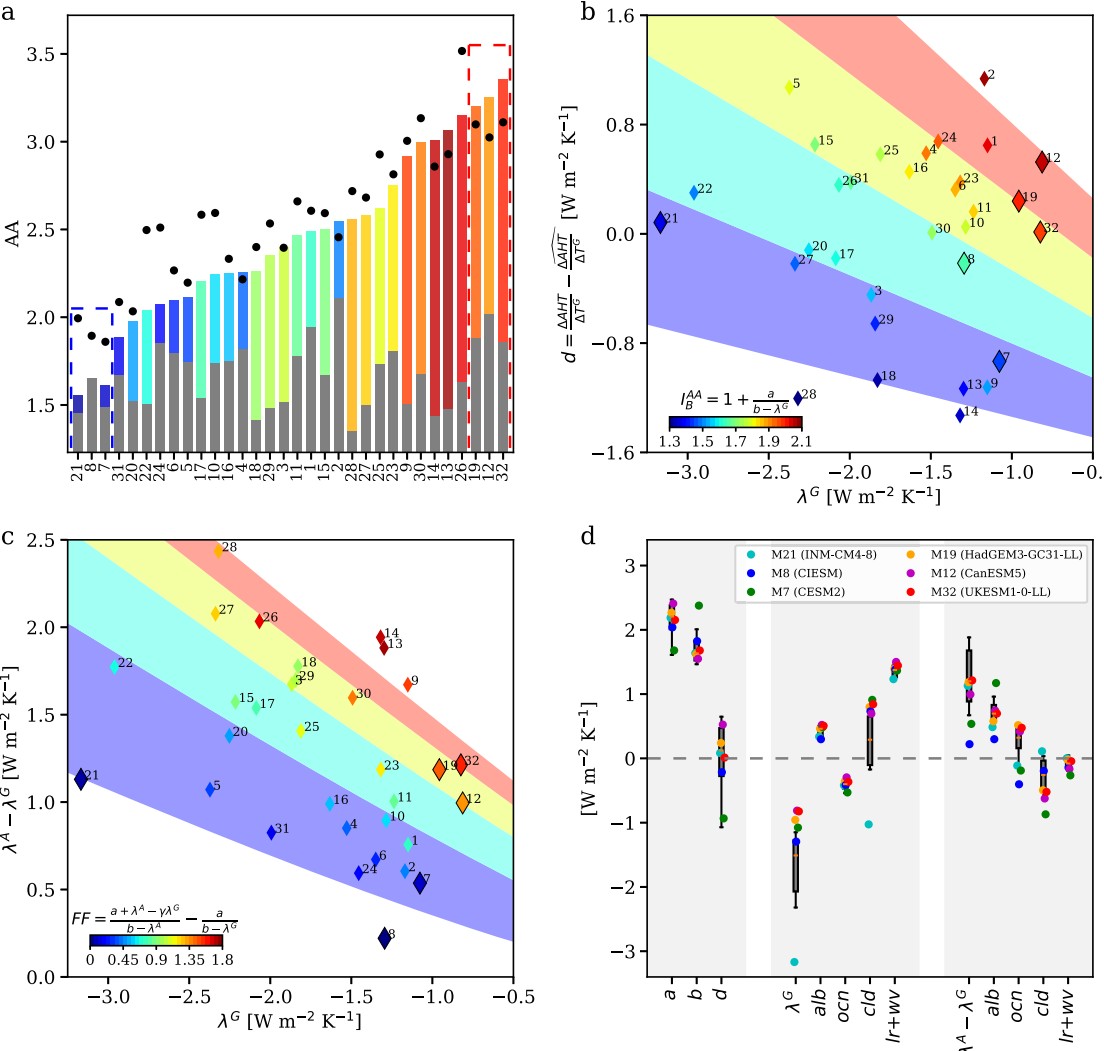

**Fig. 4 | Attribution of the spread and outliers of Arctic Amplification (AA) to physical parameters. a** The theoretical estimate, $I^{AA}$, is ranked from low to high and decomposed into the baseline AA, $I_B^{AA}$ (grey bars) and the effect of differential forcing and feedbacks, $FF$ (bars with cool-to-warm color indicating the magnitude). The degree of the model-projected AA is indicated by the black dots. The six outlier models are denoted by the blue and red boxes. The model number is shown in the x-axis. **b**. The intermodel variation of $I_B^{AA} = 1 + \frac{a}{b - \lambda^G}$ Eq. (9) explained by the variations in $\lambda^G$ and $d = \frac{\Delta AHT}{\Delta T^G} - \widehat{\frac{\Delta AHT}{\Delta T^G}}$. A higher $d$ is equivalent to a larger $a \equiv \frac{\partial AHT}{\partial T^G}$ and/or a smaller $b \equiv -\frac{\partial AHT}{\partial (T^A - T^G)}$. The magnitudes of $I_B^{AA}$ in individual models are indicated by

the cool-to-warm color. The shading reflects the value of $1 + \frac{\hat{a} + d}{b - \lambda^G}$. **c** The intermodel variations of $FF = \frac{a + \lambda^A - \gamma\lambda^G}{b - \lambda^A} - \frac{a}{b - \lambda^G}$ Eq. (10) explained by the variations in $\lambda^G$ and $\lambda^A - \lambda^G$. The magnitudes of $FF$ in individual models are indicated by the cool-to-warm color. The shading reflects the value of $\frac{\hat{a} + \lambda^A - \gamma\lambda^G}{b - \lambda^A} - \frac{\hat{a}}{b - \lambda^G}$. In (**b** and **c**), the outlier models are denoted by the enlarged diamonds. **d** The values of $a, b, d, \lambda^G, \lambda^A - \lambda^G$, and individual feedbacks in the model ensemble (boxplots indicating the 10%, 25%, mean, 75%, 90% percentiles) and in the six outlier models (colored dots). Source data are provided as a Source Data file.

the presence of the water-vapor feedback warms the Arctic more than the globe although the water-vapor feedback is weaker in the Arctic. This puzzling result can be analytically explained by our formula. Specifically, given $I^{AA} = \frac{a + b - \gamma\lambda^G}{b - \lambda^A}$, the presence of the water-vapor feedback warms the global mean by $T_{wv}^G = T^G - T_{-wv}^G = \frac{F^G}{-\lambda^G} - \frac{F^G}{-\lambda_{-wv}^G} \cong 0.31 F^G$ and warms the Arctic by $T_{wv}^A = T^A - T_{-wv}^A = T^G I^{AA} - T_{-wv}^G I_{-wv}^{AA} = \frac{F^G}{-\lambda^G}\frac{a + b - \gamma\lambda^G}{b - \lambda^A} - \frac{F^G}{-\lambda_{-wv}^G}\frac{a + b - \gamma\lambda_{-wv}^G}{b - \lambda_{-wv}^A} \cong 0.7 F^G$ (the subscript $-wv$ denotes the experiment with the water-vapor feedback set to zero). These successes highlight that, the degree of AA, emerging from controls of multiple factors, is well captured by our simple nonlinear formula.

## Discussion

Our results suggest that the degree of AA, despite governed by effects of multiple feedbacks and intricate interactions among feedbacks and

AHT, can be analytically understood. By deciphering atmospheric heat transport (AHT) and establishing a two-box EBM of AA, we show that the degree of AA is a simple nonlinear function of five key parameters as $I^{AA} \equiv 1 + \frac{a + \lambda^A - \gamma\lambda^G}{b - \lambda^A}$, where $a$ is the increasing rate of AHT with global uniform warming, $b$ is the decreasing rate of AHT with enhanced warming in the Arctic, $\gamma$ is the ratio between the Arctic and global mean radiative forcing, and $\lambda^A$ and $\lambda^G$ are the sum of climate feedbacks in the Arctic and the globe respectively. The theory conveys a concise picture of how the degree of AA emerges from essential physics as a combination of a baseline AA (which arises from the fundamental physics of AHT and exists even with uniform forcing and feedbacks) and the effect of differential forcing and feedbacks between Arctic and globe. The formula accurately captures the degree of AA in individual climate models and attributes the variation to specific physical factors that can be further targeted by modelling centers. The formula analytically

explains the degree of AA in multiple numerical studies that manipulate feedbacks and radiative forcing. More generally, the analytic theory allows us to predictively understand how the degree of AA would change if certain physical factors (e.g., feedbacks, forcing, and atmospheric diffusivity) change.

The analytic theory works beyond the existing diagnostic framework of AA which partitions the total global warming linearly into contributions of individual factors (Eqs. (3) and (4)). The diagnostic partition cannot answer how AA would change with physical factors and may misinterpret the real role of a factor (see our discussion in the introduction). Here, the analytic form provides a direct, clear understanding of how individual physical factors influence the degree of AA. The lapse-rate feedback has been widely recognized as a leading-order contributor to AA. We show here that the sum of the water-vapor and lapse-rate feedback is nearly identical between the Arctic and global mean, so the combined feedback contributes little to AA. Such compensation was not recognized by the diagnostic framework which would diagnose the combined feedback to contribute more to Arctic warming given $\Delta T^A > \Delta T^G$. The role of AHT in AA was often interpreted based on its total change $\Delta AHT$. We show here that $\Delta AHT$ consists of a forcing-like part from global-scale warming and a negative feedback part that depends on AA. Instead of by its total change, the fundamental role of AHT in AA is represented by its partial sensitivities to global uniform warming ($a$; the forcing part) and enhanced Arctic warming ($b$; the negative feedback part).

We have focused on the intermediate emission scenario SSP2-4.5, but the theory also works well for low and high emission scenarios. $I^{AA}$ is correlated with the model-projected AA at r = 0.88 for SSP1−2.6 and $r = 0.84$ For SSP5-8.5 (Fig. S3). Besides the application to understanding the intermodel variation, the theory may also be applied to understand the response of AA to changing physical parameters under paleo and future long-term climate change. Furthermore, the theory can be readily applied to the Antarctic and to understanding the hemispheric asymmetry in polar amplification. Finally, the theory provides a basis for the physical constraints on the degree of AA. Specifically, if certain parameters ($a$, $b$, $\lambda^G$, $\lambda^A$) can be observationally estimated or constrained, as shown in previous work for the global cloud feedback[48] and the Arctic summer albedo feedback[49], the uncertainty range of the degree of AA may be narrowed by substituting the observationally constrained values into the formula.

One limitation of application of the theory is that we have used a constant $\gamma$, which however could be model dependent. We note that the formula with a constant $\gamma$ tends to overestimate the high AA and underestimate the low AA in models, leading to a larger variation in AA (Fig. 4a; black dots vs. bars). One possible explanation could be that the variation in $\gamma$ may partially offset the variation in the formula with a constant $\gamma$ and potentially leads to a better match between the theory and the models.

## Methods
### Global climate models
Outputs of 32 global climate models in the Coupled Model Intercomparison Project Phase 6 (CMIP6) are used. Responses to climate change are computed from the differences in the climatology between the 1980−1995 period in the historical simulations and the 2085−2100 period in the Shared Socioeconomic Pathways (SSP) simulations. We have considered the low (SSP1−2.6), intermediate (SSP2-4.5) and high (SSP5-8.5) emission scenarios. To estimate the forced response to anthropogenic climate change, we take the mean of multiple realizations of each model, considering the effect of climate variability approximately canceled out among different realizations. The names of the 32 models and the numbers of realizations used are summarized in Table S1.

### Feedback parameters quantified by the radiative-kernel method
The method of radiative kernel is applied to calculate the feedback parameters, $\lambda_i$, as,

$$\lambda_i = \frac{\Delta R_i^{TOA}}{\Delta T}, \tag{12}$$

where $\Delta R_i^{TOA}$ is the annual-mean top-of-atmosphere (TOA) radiation anomaly induced by the change in the feedback variable and $\Delta T$ is the change in the annual-mean surface temperature. The radiative kernel contains the monthly and regionally dependent response of the TOA radiative fluxes to incremental changes in the feedback variables (e.g., albedo, air temperature, and specific humidity). $\Delta R_i^{TOA}$ is estimated by multiplying the anomaly of a feedback variable with its corresponding radiative kernel and integrating the product vertically throughout the troposphere. Specifically, the Planck feedback is computed from the change in the TOA radiation due to surface temperature changes that propagate throughout the troposphere; the lapse-rate feedback is computed from the effect of temperature departures from the vertically uniform change; the water-vapor feedback is computed from the effect of changes in air specific humidity. The albedo feedback is computed from the effect of changes in albedo, which is diagnosed from the surface downward and upward shortwave radiation. The cloud feedback is computed from the change in cloud radiative effect ($\Delta CRE$) minus the effect of non-cloud variables on $\Delta CRE$ (i.e. the difference in $\Delta R_i^{TOA}$ using the all-sky and clear-sky kernels).

### Change in atmospheric heat transport into the Arctic
The energy budget in the Arctic is influenced by the change in the atmospheric heat flux into the Arctic, termed as $\Delta AHT$. Considering the short timescale of the atmospheric energy balance, $\Delta AHT$ can be computed from the area-weighted integration of the atmospheric energy convergence (i.e., the difference between the net surface and TOA energy fluxes) as,

$$\Delta AHT = -\iint_{Arctic} \Delta \left( R^{TOA} - R^{SFC} + LH + SH \right) dA. \tag{13}$$

$\Delta AHT$ is formulated as a function of the changes in global mean surface temperature $\Delta T^G$ and meridional temperature gradient $\Delta T^A - \Delta T^G$ as

$$\Delta AHT \cong a\Delta T^G - b\left( \Delta T^A - \Delta T^G \right). \tag{14}$$

A reformulation of Eq. (14) gives

$$\frac{\Delta AHT}{\Delta T^G} \cong a - b(AA - 1). \tag{15}$$

This formula is supported by the significant correlation between $\frac{\Delta AHT}{\Delta T^G}$ and $AA - 1$ across models (Fig. 2a; r = -0.70). The intermodel regression estimate $\hat{a} \cong 2.1$ W m$^{-2}$ K$^{-1}$ and $\hat{b} \cong 1.7$ W m$^{-2}$ K$^{-1}$. The hat symbol denotes a constant estimate that works generally for the model ensemble instead of individual models. By using $\hat{a}$ and $\hat{b}$, Eq. (14) reasonably captures $\Delta AHT^A$ in individual models (r = 0.71), capturing -50% of the variance in $\Delta AHT$ among models (Fig. S1). The deviation of $\Delta AHT$ from this estimate with constant $\hat{a}$ and $\hat{b}$ reflects the variations of $a$ and $b$ among models. It is however challenging to accurately estimate $a$ and $b$ for individual models. The spread of the climate responses among different SSPs (SSP1−2.6, SSP2-4.5, SSP5-8.5) do not always yield a reasonable regression line between $\frac{\Delta AHT}{\Delta T^G}$ and AA−1 for estimating $a$ and $b$ (Fig. S4). This cloud be due to either the varied $a$ and $b$ among different SSPs or insufficient realizations to cleanly remove the contamination from internal variability.

We approximately estimate $a$ and $b$ in individual models as deviations from $\hat{a}$ and $\hat{b}$ so that Eq. (15) reproduces the model-projected $\frac{\Delta AHT}{\Delta T^G}$. Specifically, the difference between the model-projected $\frac{\Delta AHT}{\Delta T^G}$ and the prior estimate $\widehat{\frac{\Delta AHT}{\Delta T^G}} = \hat{a} - \hat{b}(AA-1)$, that is, $d = \frac{\Delta AHT}{\Delta T^G} - \widehat{\frac{\Delta AHT}{\Delta T^G}}$ is attributed to $a$ and $b$ as

$$a = \hat{a} + (1-\delta)d, \qquad (16)$$

$$b = \hat{b} - \delta d/(AA-1). \qquad (17)$$

In the main text, we use $\delta = 0.5$, that is, attributing $d$ equally to $a$ and $b$. With the model-dependent $a$ and $b$, Eq. (14) fully reproduces $\Delta AHT$ (Fig. S1).

We show below that the value of $I^{AA}$ is insensitive to the exact attribution of $d = \frac{\Delta AHT}{\Delta T^G} - \widehat{\frac{\Delta AHT}{\Delta T^G}}$ to $a$ and $b$.

If $d$ is all attributed to $a$, $a = \hat{a} + d$ and $b = \hat{b}$, then we have,

$$I^{AA} - 1 \equiv \frac{\hat{a} + d + \lambda^A - \gamma\lambda^G}{\hat{b} - \lambda^A}. \qquad (18)$$

If $d$ is all attributed to $b$, $a = \hat{a}$ and $b = \hat{b} - d/(AA-1)$, then we have,

$$
\begin{aligned}
I^{AA} - 1 &\equiv \frac{\hat{a} + \lambda^A - \gamma\lambda^G}{\hat{b} - \frac{d}{AA-1} - \lambda^A} \\
&\cong \frac{\hat{a} + \lambda^A - \gamma\lambda^G}{\hat{b} - \lambda^A} \left(1 + \frac{d}{(AA-1)(\hat{b}-\lambda^A)} + \frac{d^2}{(AA-1)^2 (\hat{b}-\lambda^A)^2} \right.\\
&\qquad \left. + \cdots \frac{d^n}{(AA-1)^n (\hat{b}-\lambda^A)^n} \right) \\
&\cong \frac{\hat{a} + \lambda^A - \gamma\lambda^G + d}{\hat{b} - \lambda^A} - \frac{d^2}{(AA-1)(\hat{b}-\lambda^A)^2} + \frac{d^2}{(AA-1)(\hat{b}-\lambda^A)^2}\frac{b-\lambda^A}{\hat{b}-\lambda^A} \\
&\qquad + \cdots \frac{d^n}{(AA-1)^{n-1}(\hat{b}-\lambda^A)^n}\frac{b-\lambda^A}{\hat{b}-\lambda^A} \\
&\cong \frac{\hat{a} + \lambda^A - \gamma\lambda^G + d}{\hat{b} - \lambda^A} - \frac{d^{n+1}}{(AA-1)^n(\hat{b}-\lambda^A)^{n+1}} \\
&\cong \frac{\hat{a} + \lambda^A - \gamma\lambda^G + d}{\hat{b} - \lambda^A} - (AA-1)\left(\frac{\hat{b}-b}{\hat{b}-\lambda^A}\right)^{n+1} \cong \frac{\hat{a} + \lambda^A - \gamma\lambda^G + d}{\hat{b} - \lambda^A}.
\end{aligned} \qquad (19)
$$

The above derivation uses the relation $AA - 1 \cong I^{AA} - 1 = \frac{\hat{a} + \lambda^A - \gamma\lambda^G}{\hat{b} - \lambda^A}$ and assumes the deviation of b from $\hat{b}$ $\left|\hat{b} - b\right| < \hat{b} - \lambda^A \cong 2.1$ W m$^{-2}$ K$^{-1}$.

So $I^{AA}$ is expected to be nearly identical when we attribute $d = \frac{\Delta AHT}{\Delta T^G} - \widehat{\frac{\Delta AHT}{\Delta T^G}}$ all to $a$ or $b$. Indeed, $I^{AA}$ is largely invariant with either attribution and consistently correlated with the model-projected AA at $r = 0.91$ (Fig. S5).

The invariance of $I^{AA}$ with the attribution can be mathematically derived by considering arbitrary attribution, $a = \hat{a} + a'$, $b = \hat{b} + b'$ and their relation $a' - b'(AA-1) = d$. Let $\hat{\alpha} = \hat{a} + \lambda^A - \gamma\lambda^G$, $\hat{\beta} = \hat{b} - \lambda^A$, $\alpha = a + \lambda^A - \gamma\lambda^G$, $\beta = b - \lambda^A$, we have

$$
\begin{aligned}
I^{AA} - 1 &\equiv \frac{\hat{\alpha} + a'}{\hat{\beta} + b'} \cong \left(\frac{\hat{\alpha}}{\hat{\beta}} + \frac{d + b'(AA-1)}{\hat{\beta}}\right)\left(1 - \frac{b'}{\hat{\beta}}\right) \\
&= \frac{\hat{\alpha} + d}{\hat{\beta}} + \frac{b'(AA-1)}{\hat{\beta}} - \frac{\hat{\alpha}b'}{\hat{\beta}^2} - \frac{db'}{\hat{\beta}^2} - \frac{b'^2}{\hat{\beta}^2}(AA-1)
\end{aligned}
$$

Substituting $\frac{\hat{\alpha}}{\hat{\beta}}(1 + \frac{a'}{\alpha} - \frac{b'}{\beta})$ leads to $AA - 1 \cong I^{AA} - 1 = \frac{\hat{\alpha}+a'}{\hat{\beta}+b'} = \frac{\hat{\alpha}}{\hat{\beta}}\left(1 + \frac{a'}{\alpha}\right)\left(1 - \frac{b'}{\beta}\right) =$

$$
\begin{aligned}
I^{AA} - 1 &\cong \frac{\hat{\alpha}+d}{\hat{\beta}} + \frac{b'}{\hat{\beta}}\frac{\hat{\alpha}}{\hat{\beta}}\left(1 + \frac{a'}{\alpha} - \frac{b'}{\beta}\right) - \frac{\hat{\alpha}b'^2}{\hat{\beta}^2} - \frac{db'}{\hat{\beta}^2} - \frac{b'^2}{\hat{\beta}^2}(AA-1) \\
&= \frac{\hat{\alpha}+d}{\hat{\beta}} + \frac{b'}{\hat{\beta}^2}\frac{a'}{} - \frac{\hat{\alpha}b'^2}{\hat{\beta}^3} - \frac{db'}{\hat{\beta}^2} - \frac{b'^2}{\hat{\beta}^2}(AA-1) \\
&= \frac{\hat{\alpha}+d}{\hat{\beta}} + \frac{b'}{\hat{\beta}^2}\left(d + b'(AA-1)\right) - \frac{\hat{\alpha}b'^2}{\hat{\beta}^3} - \frac{db'}{\hat{\beta}^2} - \frac{b'^2}{\hat{\beta}^2}(AA-1) = \frac{\hat{\alpha}+d}{\hat{\beta}} - \frac{\hat{\alpha}b'^2}{\hat{\beta}^3} \\
&\cong \frac{\hat{\alpha}+d}{\hat{\beta}} = \frac{\hat{a}+d+\lambda^A-\gamma\lambda^G}{\hat{b}-\lambda^A}.
\end{aligned} \qquad (20)
$$

**Theoretical derivation of the formula of $\Delta AHT$ Eq. (5)**

The formula of $\Delta AHT \cong a\Delta T^G - b\left(\Delta T^A - \Delta T^G\right)$ can also be theoretically derived from the diffusive formulation of AHT. Specifically, according to the diffusive theory, AHT into the Arctic can be written as,

$$\text{AHT} = -\beta D_T C_p \frac{\partial T}{\partial \theta}\Big|_{\theta_o} - \beta D_q L_v \frac{\partial q}{\partial \theta}\Big|_{\theta_o}, \qquad (21)$$

where $T$ and $q$ are surface temperature and specific humidity respectively, $D_q$ and $D_T$ are the diffusivity for moisture and temperature respectively, $\beta = \frac{\cos\theta_o}{1-\sin\theta_o}\frac{P_s}{R^2 g}$ scales AHT to the unit of W m$^{-2}$ K$^{-1}$, $P_s$ is the surface pressure, $R$ is the earth radius, $\theta$ is the latitude radian, and $\theta_o = 65°$N defines the Arctic boundary, and $|_{\theta_o}$ indicates values around $\theta_o$. Under climate change, $\frac{\partial q}{\partial\theta}|_{\theta_o}$ and $\frac{\partial T}{\partial\theta}|_{\theta_o}$ changes as,

$$\Delta\frac{\partial T}{\partial\theta}\Big|_{\theta_o} = \zeta\left(\Delta T^A - \Delta T^G\right), \qquad (22)$$

$$
\begin{aligned}
\Delta\frac{\partial q}{\partial\theta}\Big|_{\theta_o} &= \frac{\partial\left(\frac{dq}{dT}\right)}{\partial\theta}\Big|_{\theta_o}\Delta T^G + \widetilde{\frac{dq}{dT}}\Big|_{\theta_o}\frac{\partial\Delta T}{\partial\theta}\Big|_{\theta_o} = \frac{L_v}{R_v T^2}\frac{\partial q}{\partial\theta}\Big|_{\theta_o}\Delta T^G \\
&\quad + \frac{L_v}{R_v T^2}\widetilde{q}\Big|_{\theta_o}\zeta\left(\Delta T^A - \Delta T^G\right),
\end{aligned} \qquad (23)
$$

where the tilde symbol $\tilde{\ }$ indicates the warming-weighted mean and $\zeta \equiv \frac{\frac{\partial T}{\partial\theta}|_{\theta_o}}{\Delta T^A - \Delta T^G}$. In Eq. (23), we have used the relation $\frac{dq}{dT} = \frac{L_v q}{R_v T^2}$, which is derived from the Clausius-Clapeyron relation and the assumption that relative humidity stays roughly constant under climate change. Substituting Eqs. (22) and (23) into Eq. (21) leads to the same formula of $\Delta AHT$ as Eq. (5),

$$\Delta\text{AHT} = a\Delta T^G - b\left(\Delta T^A - \Delta T^G\right), \qquad (24)$$

with

$$a \equiv \frac{\partial AHT}{\partial T^G} = -\beta D_q\frac{L_v^2}{R_v T^2}\frac{\partial q}{\partial\theta}\Big|_{\theta_o}, \qquad (25)$$

$$b \equiv \frac{\partial AHT}{\partial\left(T^A - T^G\right)} = \beta\zeta\left(D_q\frac{L_v^2}{R_v T^2}\widetilde{q}\Big|_{\theta_o} + D_T C_p\right). \qquad (26)$$

The above derivation shows that the formula $\Delta AHT = a\Delta T^G - b(\Delta T^A - \Delta T^G)$ can be theoretically derived from the diffusive formulation of AHT and the parameter $a$ and $b$ are functions of basic parameters of the climate system.

Approximate values of $a$ and $b$ ca be estimated using Eqs. (25) and (26). According to the climatological distribution of surface (2 m) specific humidity (Fig. S6a), $\frac{\partial q}{\partial\theta}|_{\theta_o} \cong 7$g kg$^{-1}$,

$\tilde{q}|_{\theta_o} \cong 2\,\mathrm{g\,kg^{-1}}$. According to the meridional structure of the warming pattern (Fig. S6b), $\zeta \equiv \frac{\frac{\partial T}{\partial \theta}|_{\theta_o}}{\Delta T^A - \Delta T^G} \cong 1$. For diffusivity, Ref. 50 estimated $D_q \cong 1.8$ km$^2$ s$^{-1}$ and $D_T \cong 1.2$ km$^2$ s$^{-1}$ based on an aquaplanet simulation, while Ref. 51 used a fixed diffusivity $D_{T,q} \cong 1.06$ km$^2$ s$^{-1}$ that best fits their results. The estimates from the intermodel regression, $\hat{a} \cong 2.1\,\mathrm{W\,m^{-2}\,K^{-1}}$ and $\hat{b} \cong 1.7\,\mathrm{W\,m^{-2}\,K^{-1}}$, are reproduced by Eqs. (25) and (26) if $D_q \cong 1.6$ km$^2$ s$^{-1}$ and $D_T \cong 1.0$ km$^2$ s$^{-1}$. We note that the ratio between $a$ and $b$ is a function of the climatological distribution of specific humidity and $D_T/D_q$ as

$$\frac{a}{b} = \frac{\frac{\partial q}{\partial \theta}|_{\theta_o}}{\tilde{q}|_{\theta_o} + D_T C_p R_v T^2 / D_q L_v^2} \cong \frac{7}{2 + 5.7 D_T/D_q}, \qquad (27)$$

whose value should be around (or more likely slightly above) 1.

## Ocean feedback

The energy budgets in the Arctic and the globe are influenced by the changes in the ocean heat flux, termed as $\Delta O$. Considering the short timescale of the ocean-mixed-layer energy balance, $\Delta O$ can be estimated from the change in the net surface fluxes as,

$$\Delta O = \Delta(SH + LH - R^{SFC}), \qquad (28)$$

where $SH$ and $LH$ are the surface (upward) sensible and latent heat flux respectively and $R^{SFC}$ is the net downward radiation flux at the surface. To facilitate the derivation and simplify interpretation, we have written $\Delta O$ as a feedback,

$$\Delta O = \lambda_O \Delta T, \qquad (29)$$

where $\lambda_O$ is diagnosed as $\Delta O/\Delta T$ for the globe and the Arctic. While this is just a mathematical treatment, it can be physically justified as $\Delta O$ is approximately proportional to $\Delta T$ in individual models (Fig. S2). For the globe, $\lambda_o^G = -0.41 \pm 0.10$ W m$^{-2}$ K$^{-1}$; For the Arctic, $\lambda_o^A = -0.14 \pm 0.37$ W m$^{-2}$ K$^{-1}$, where $\pm$ indicates the s.d. among models. We note that $\lambda_O$ is similar to the ocean heat exchange parameter $\kappa$ in ref. 42 but subtly different from (and thus smaller in magnitude than) the parameter $\eta$ in Ref. 52 which further considered a deep-ocean layer.

## Decompose the degree of AA into contributions of individual physical factors

$I^{AA}$ is decomposed into a baseline AA and the net effect of differential radiative forcing and feedbacks between Arctic and globe, that is,

$$I^{AA} = 1 + \frac{a + \lambda^A - \hat{\gamma}\lambda^G}{b - \lambda^A} = \underbrace{1 + \frac{a}{b - \lambda^G}}_{\substack{\text{baseline AA} \\ I_B^{AA}}} + \underbrace{\frac{a + \lambda^A - \hat{\gamma}\lambda^G}{b - \lambda^A} - \frac{a}{b - \lambda^G}}_{\substack{\text{effect of differential forcing and feedbacks} \\ FF}}. \qquad (30)$$

The effects of differential forcing and feedbacks, $FF$, can be further separated. However, we should note that, due to the nonlinear format of $FF$, the separate effects of differential forcing and feedbacks depend on each other. For example, the effect of differential forcing may be estimated as $\frac{(1-\hat{\gamma})\lambda^G}{b - \lambda^G}$ or $\frac{(1-\hat{\gamma})\lambda^G}{b - \lambda^A}$ depending on whether the effect of differential feedbacks has already been included. Both estimates are negative but they differ by -0.1. In Fig. 2e, the effect of differential forcing is computed with the effect of feedbacks already included as $\frac{(1-\hat{\gamma})\lambda^G}{b - \lambda^A}$.

The effect of differential feedbacks is estimated by replacing the total Arctic feedback with the total global feedback, that is,

$$\frac{a + \lambda^A - \hat{\gamma}\lambda^G}{b - \lambda^A} - \frac{a + \lambda^G - \hat{\gamma}\lambda^G}{b - \lambda^G}. \qquad (31)$$

The contribution of individual feedback $\lambda_i$ is then estimated by replacing its Arctic value with its global value, that is,

$$\frac{a + \lambda^A - \hat{\gamma}\lambda^G}{b - \lambda^A} - \frac{a + \lambda^{A'} + \hat{\gamma}\lambda^G}{b - \lambda^{A'}}. \qquad (32)$$

with $\lambda^{A'} = \lambda^A - \lambda_i^A + \lambda_i^G$.

## Data availability
The CMIP6 outputs are available from the Earth System Grid Federation (ESGF) Portal at https://esgf-node.llnl.gov/search/cmip6/. The source data underlying the main figures is available at https://zenodo.org/records/10976607. Source data are provided with this paper.

## Code availability
The code for the radiative-kernel method is available from Professor Yi Huang's group page https://data.mendeley.com/datasets/3drx8fmmz9/1. The script for analyses and generating figures is available at https://zenodo.org/records/10976607.

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

## Acknowledgements

This study was supported by Office of Science, U.S. Department of Energy Biological and Environmental Research as part of the Regional and Global Model Analysis program area. The Pacific Northwest National Laboratory (PNNL) is operated for DOE by Battelle Memorial Institute under contract DE-AC05-76RLO1830. We acknowledge the WCRP Working Group on Coupled Modeling, which is responsible for CMIP, and the climate modeling groups for producing and making available their model outputs. This research used resources of the National Energy Research Scientific Computing Center, which is supported by the Office of Science of the U.S. Department of Energy under Contract No. DE-AC02-05CH1123. We also would like to acknowledge the data access and computing support provided by the NCAR CMIP Analysis Platform (https://doi.org/10.5065/D60R9MSP).

## Author contributions

W.Z. designed the research and conducted the analysis. R.L., S.X. and J.L. contributed to improving the analyses and interpretation. W.Z. drafted the manuscript and all the authors edited the paper.

## Competing interests

The authors declare no competing interests.
