## [Peer Review File · Nature Communications]

An analytic theory for the degree of Arctic AmplificationREVIEWER COMMENTS

Reviewer #1 (Remarks to the Author):

The authors develop a theoretical framework to quantitatively link the strength of Arctic and non-Arctic/global forcing and feedbacks, the response of atmospheric heat transport to these and the resulting Arctic amplification of climate change.

The idea is appealing and some of the results are promising, but I have reservations on several of the lines of evidence suggested in the manuscript.

1) Derivation of \hat{a} and \hat{b} : The authors state “The theoretical diffusive formula of AHT estimates the same values of \hat{a} and \hat{b} under a constant diffusivity of $D= 106 \text{ m}^2 \text{ s}^{-1}$ (Methods)”

This sounds like an independent way of estimating the parameters on a theoretical basis, but reading the corresponding parts of the methods section, where “typical” values of large-scale climate state variables are used without further justification, suggests that these might just have been chosen to deliver the desired result. I suggest to underpin these calculations by using climate model output to estimate the values needed for the theoretical derivation, or to omit this line of evidence.

2) Derivation of a and b (model-specific): These are not actually derived using the underlying regression for each parameter, but just fitted by assuming that each parameter accounts for half the deviation from the inter-model mean parameters. The fact that this works equally well when attributing all the deviation from the mean parameter to just one of either a or b might indicate arbitrary overfitting rather than robustness of the method.

I do not understand why there are not enough climate states to derive these parameters individually – one could use PI-control, different 30-year intervals from historical, 1pct CO₂ scenario or 4xCO₂ runs as well. If that does not work out, maybe there is a problem with the method.

I was further wondering if a should be temperature-dependent and could get a more theoretical foundation based on the Clausius-Clapeyron relation, and whether the ‘global’ parameter should instead be sub-Arctic. Given the small surface area of the Arctic, I do not expect the latter to make a strong quantitative difference, but it would seem more in line with the conceptual sketch given in Fig1.

Reviewer #2 (Remarks to the Author):

This is a very interesting exercise looking at what factors impact Arctic Amplification (AA) in models. The article is worth publishing. Two major comments: this is in no way a 'predictive' theory of AA. It only amounts to 'diagnostics' of AA. If the authors want to keep 'predictive' in the title and call it a predictive theory in the article body, I will reject it. The conceptual model does not include any physical basis for how the increase in AHT happens in a warmer climate. My second comment is about the absence of time in the analysis. It assumes everything is in equilibrium while the models are definitely not. But

overall, the analysis is very interesting and seems to be very solid.

Reviewer #3 (Remarks to the Author):

Arctic amplification is an aspect of climate change that has widespread interest and has been the subject of many lines of inquiry concerning its mechanisms. This manuscript offers a new theoretical framework to the factors governing Arctic amplification that is novel and grounded in a fundamental aspect of how the atmosphere transports heat to the pole. It's a very illuminating new approach and I recommend publication upon a minor revision.

The manuscript deploys a well justified approach to atmospheric heat transport (AHT) to Arctic Amplification (AA). AHT's changes with climate can be thought of as depending on two mean-state quantities: the global-mean surface temperature, which increases the heat transport, and the temperature contrast in latitude, AA decreases this which is an offsetting factor. This is combined with imposed regional feedbacks and forcing to capture the behavior of comprehensive climate models.

Two overarching points that should be acknowledged in the manuscript:

A. The lapse rate feedback is prescribed from the results of GCMs: the Arctic lapse rate is affected by a combination of factors, including radiative forcing and atmospheric heat transport. The theoretical understanding of this is clear about why different forcing and feedback have different lapse rate changes (e.g., Cronin and Jansen 2016), and there are feedback analysis approaches that are designed to separate this (e.g., Feldl et al 2020). So, in principle, the Arctic box's feedback depends on how its solution evolves. The empirical success of the authors' approach suggests this isn't a leading-order effect---good results despite the omission of this complicating factor. But it's important to communicate this aspect of the new framework introduced.

B. The need to impose feedbacks from climate model simulation means this framework in practice is still heavily diagnostic (vs. purely diagnostic approaches described near L96). This means the new framework is not predictive, despite the title and abstract's language.

Some clarifications:

1. The theory's "baseline AA" (L178) has a similar dependence on the global feedback parameter as moist energy balance model theory (Merlis and Henry 2018)---it appears in the denominator. Are these the same? Is the value of 1.67 (L181) the same?

2. Fig. 2e shows the baseline AA is the largest factor, larger than the feedback part (gray vs. yellow). Is it fair to conclude that the energy transport related part of AA is dominant? In other words, in the

diagnostic approaches that are critiqued in the introduction (Pithan & Mauritsen 2014), the AHT term is not dominant but one can't make firm conclusions based on that approach. Can one now make a firm conclusion on that basic question?

Minor presentation revisions:

- The diagram in Fig. 1 shows two regions: Arctic and everywhere else, but the equations suggest the feedback parameter for "everywhere else" is actually the global mean. Is there this double counting where the diagnosed Arctic feedback enters both λ_A and G ?

- L124 typo before which- paragraph ending L210: this sounds like a rehash of the central discussion of Held & Shell 2012 who suggest a different feedback decomposition that assumes constant relative humidity as the reference response, but applied to the zonal-mean. But others have presented zonal-mean feedbacks using that different decomposition, so I don't think this is particularly new, and maybe not worth emphasizing strongly particularly given the ambiguity of what the Arctic lapse rate is controlled by. For example, Hahn et al. 2021 did the diagnostic energy budget approach with both feedback decompositions and found: "As a result, the relative contribution of the lapse-rate feedback to Arctic amplification is weakened in the fixed-RH framework, with stronger contributions from the albedo feedback and poleward moisture transport." Using all of Fig. 3 on this point is overkill to me.

- L272 typo with λ superscript, repeated G

- L293 highlighting individual models by this coding is not the most effective way to communicate, perhaps point to the right side of Fig. 4 columns instead

- L376: does this ocean heat uptake parameter agree with previous published results like Geoffroy et al. 2013 and work by J. Gregory? I had a ~50% more negative number in mind

- eqn. S9, S10: there's never a saturation specific humidity defined or relative humidity stated that it is assumed constant

- Fig. 4 caption uses FF, but that's not defined in the main text

Response to Reviewer #1:

The authors develop a theoretical framework to quantitatively link the strength of Arctic and non-Arctic/global forcing and feedbacks, the response of atmospheric heat transport to these and the resulting Arctic amplification of climate change. The idea is appealing and some of the results are promising, but I have reservations on several of the lines of evidence suggested in the manuscript.

Thank you for your insightful review. Following your suggestions, we have used climate model outputs to estimate the values of a and b in the theoretical derivation and explained why our formula works well independent of the exact attribution of $d = \frac{\Delta AHT}{\Delta T^G} - \frac{\widehat{\Delta AHT}}{\Delta T^G}$ to a and b . We have provided point-by-point responses to your comments below.

1) Derivation of \hat{a} and \hat{b} : The authors state “The theoretical diffusive formula of AHT estimates the same values of \hat{a} and \hat{b} under a constant diffusivity of $D= 106 \text{ m}^2 \text{ s}^{-1}$ (Methods)” This sounds like an independent way of estimating the parameters on a theoretical basis, but reading the corresponding parts of the methods section, where “typical” values of large-scale climate state variables are used without further justification, suggests that these might just have been chosen to deliver the desired result. I suggest to underpin these calculations by using climate model output to estimate the values needed for the theoretical derivation, or to omit this line of evidence.

Following your suggestion, we have used the climate model output to estimate the values needed for the theoretical estimate of a and b . We have rewritten the derivation so it is more rigorous (see the Method section). Specifically, we have used the partial derivative to latitude radian $\frac{\partial}{\partial \theta}$ instead of δ , and considered the different diffusivities for temperature and moisture.

We would like to note that the purpose of this theoretical derivation is not to exactly reproduce the values of \hat{a} and \hat{b} estimated from the intermodel regression, but to show that $\Delta AHT = a\Delta T^G - b(\Delta T^A - \Delta T^G)$ can be theoretically derived from the diffusive formula of AHT, and that a and b are functions of basic parameters of the climate systems (Eq. S14 and S15).

2) Derivation of a and b (model-specific): These are not actually derived using the underlying regression for each parameter, but just fitted by assuming that each parameter accounts for half

the deviation from the inter-model mean parameters. The fact that this works equally well when attributing all the deviation from the mean parameter to just one of either a or b might indicate arbitrary overfitting rather than robustness of the method.

If a model presents a higher ΔAHT than the prior estimate $\widehat{\Delta AHT} = \hat{a} - \hat{b}(AA - 1)$, this model presents a higher a (than \hat{a}) and/or a lower b (than \hat{b}), either a higher a or a lower b would increase the value of $I^{AA} \equiv 1 + \frac{a + \lambda^A - \gamma \lambda^G}{b - \lambda^A}$. **The fact that alternative attributions work robustly can be mathematically derived as follows.**

If $d = \frac{\Delta AHT}{\Delta T^G} - \frac{\widehat{\Delta AHT}}{\Delta T^G}$ is all attributed to a , that is, $a = \hat{a} + d$ and $b = \hat{b}$, we have

$$I^{AA} \equiv 1 + \frac{\hat{a} + \lambda^A - \gamma \lambda^G}{\hat{b} - \lambda^A} + \frac{d}{\hat{b} - \lambda^A}$$

If d is all attributed to b , that is, $a = \hat{a}$ and $b = \hat{b} - d/(AA - 1)$, we have,

$$\begin{aligned} I^{AA} - 1 &\equiv \frac{\hat{a} + \lambda^A - \gamma \lambda^G}{\hat{b} - \frac{d}{AA - 1} - \lambda^A} \\ &\cong \frac{\hat{a} + \lambda^A - \gamma \lambda^G}{\hat{b} - \lambda^A} \left(1 + \frac{d}{(AA - 1)(\hat{b} - \lambda^A)} + \frac{d^2}{(AA - 1)^2(\hat{b} - \lambda^A)^2} + \dots \right. \\ &\quad \left. + \frac{d^n}{(AA - 1)^n(\hat{b} - \lambda^A)^n} \right) \\ &\cong \frac{\hat{a} + \lambda^A - \gamma \lambda^G + d}{\hat{b} - \lambda^A} - \frac{d^2}{(AA - 1)(\hat{b} - \lambda^A)^2} + \frac{d^2}{(AA - 1)(\hat{b} - \lambda^A)^2} \frac{b - \lambda^A}{\hat{b} - \lambda^A} \\ &\quad + \dots \frac{d^n}{(AA - 1)^{n-1}(\hat{b} - \lambda^A)^n} \frac{b - \lambda^A}{\hat{b} - \lambda^A} \\ &\cong \frac{\hat{a} + \lambda^A - \gamma \lambda^G + d}{\hat{b} - \lambda^A} - \frac{d^{n+1}}{(AA - 1)^n(\hat{b} - \lambda^A)^{n+1}} \\ &\cong \frac{\hat{a} + \lambda^A - \gamma \lambda^G + d}{\hat{b} - \lambda^A} - (AA - 1) \left(\frac{\hat{b} - b}{\hat{b} - \lambda^A} \right)^{n+1} \cong \frac{\hat{a} + \lambda^A - \gamma \lambda^G + d}{\hat{b} - \lambda^A} \end{aligned}$$

The above derivation uses the relation $AA - 1 \cong I^{AA} - 1 = \frac{\hat{a} + \lambda^A - \gamma \lambda^G}{b - \lambda^A}$ and assumes the deviation of b from \hat{b} $|\hat{b} - b| < \hat{b} - \lambda^A \cong 2.1 \text{ W m}^{-2} \text{ K}^{-1}$.

The above results indicate that the value of I^{AA} is rather insensitive to the exact attribution of

$d = \frac{\Delta AHT}{\Delta T^G} - \frac{\widehat{\Delta AHT}}{\Delta T^G}$ to a and b . In Methods, we further derive the insensitivity of I^{AA} to attribution by considering an arbitrary attribution of d to a and b .

Given $I^{AA} = 1 + \frac{a+\lambda^A-\gamma\lambda^G}{b-\lambda^A} \cong 1 + \frac{\hat{a}+\lambda^A-\gamma\lambda^G}{\hat{b}-\lambda^A} + \frac{d}{\hat{b}-\lambda^A}$, we can now interpret the effect of AHT from \hat{a} , \hat{b} and $d = \frac{\Delta AHT}{\Delta T^G} - \frac{\widehat{\Delta AHT}}{\Delta T^G}$. This is arguably simpler than understanding the effect of AHT from a and b . Also, it shows more clearly why directly using ΔAHT to understand the effect of AHT is not the best way. We have added this new insight into the manuscript (L228-236).

I do not understand why there are not enough climate states to derive these parameters individually – one could use PI-control, different 30-year intervals from historical, 1pct CO2, scenario or 4xCO2 runs as well. If that does not work out, maybe there is a problem with the method. I was further wondering if a should be temperature-dependent and could get a more theoretical foundation based on the Clausius-Clapeyron relation, and whether the ‘global’ parameter should instead be sub-Arctic. Given the small surface area of the Arctic, I do not expect the latter to make a strong quantitative difference, but it would seem more in line with the conceptual sketch given in Fig1.

The underlying assumption that one can estimate a and b from the spread of different climate responses is that a and b are approximately fixed among these warming experiments. We have replotted Fig. 2a using different scenario simulations --SSP126, SSP245 and SSP585 (Fig. R1). For some models, there is not a notable regression line among SSPs; and for some models, the regression line among SSPs indicates spurious negative a and/or b . This could be due to either the varied a and b among different SSPs or insufficient realizations to cleanly remove the contamination from internal variability (see Table S1; even fewer realizations are available for the 1pctCO2 and 4xCO2 experiments).

Fortunately, as demonstrated above, the value of I^{AA} is insensitive to the exact attribution of $d = \frac{\Delta AHT}{\Delta T^G} - \frac{\widehat{\Delta AHT}}{\Delta T^G}$ to a and b so our results are not affected.

It would be interesting to understand if/how the parameters of AA vary under different warming scenarios, and moreover if they present certain temperature dependency. However, since the focus of this study is to establish the theory and considering the current difficulty in accurately

estimating a and b for individual models, it may be better to leave this subject for future work.

We use “globe” (following Eq. 1 and 2) as one can drop the term AHT in the global mean energy balance but not for the sub-Arctic. Using “globe” makes our derivation simpler and naturally connects to the definition of AA (the ratio between the Arctic mean and the global mean). We have modified the conceptual sketch to avoid inconsistency.

Fig. R1: Scatterplot between $AA-1$ and the change in atmospheric heat transport into the Arctic normalized by the global mean warming ($\Delta AHT/\Delta T^G$) across models. The SSP126, SSP245 and SSP585 warming scenarios are considered and shown in symbols. The results of SSPs of each model is connected by a line.

Response to Reviewer #2:

This is a very interesting exercise looking at what factors impact Arctic Amplification (AA) in models. The article is worth publishing. Two major comments: this is in no way a 'predictive' theory of AA. It only amounts to 'diagnostics' of AA. If the authors want to keep 'predictive' in the title and call it a predictive theory in the article body, I will reject it. The conceptual model does not include any physical basis for how the increase in AHT happens in a warmer climate. My second comment is about the absence of time in the analysis. It assumes everything is in equilibrium while the models are definitely not. But overall, the analysis is very interesting and seems to be very solid.

Thank you for your insightful review.

To your first comment:

Following your comment, we have removed the word “predictive” from the title and the article body, and explained more clearly what new insights our study provides beyond the previous diagnostic framework.

The reason we use “predictive” initially is that we now have a formula that estimates the degree of AA from its physical factors and can “predictively” understand how AA would change with the physical factors. We think this is beyond the general diagnostic framework, which can only tell us how much a factor contributes to the total Arctic or global warming. But we agree that the formula is not a forecast model that can directly predict the future degree of AA.

We do not fully understand this sentence – “The conceptual model does not include any physical basis for how the increase in AHT happens in a warmer climate”. **A key contribution of this work is to reveal the intricate role of AHT in AA.** The change in AHT is a combination of both forcing and feedback. For forcing, AHT increases as global-scale warming increases the meridional moisture gradient (de^*/dT is higher in the warmer tropics) and contributes to AA. For negative feedback, AHT decreases as enhanced Arctic warming reduces the meridional temperature gradient and dampens AA. **In combination, AHT can either increase or decrease in a warmer climate (see Fig. 2a).** Previous studies have directly used ΔAHT to talk about the contribution of AHT to AA. ΔAHT , however, is a combination of forcing and feedback that depends on AA itself.

Here, by formulating $\Delta AHT \cong a\Delta T^G - b(\Delta T^A - \Delta T^G)$ and $AA \equiv \frac{\Delta T^A}{\Delta T^G} = 1 + \frac{a + \lambda^A - \gamma\lambda^G}{b - \lambda^A}$, we show

that the role of AHT in AA is represented by two key parameters: a – the increasing rate of AHT with global uniform warming, and b – the decreasing rate of AHT with enhanced Arctic warming.

To your second comment:

Our investigation of AA is based on the difference between 2085-2100 and 1980-1995. The derivation is based on the energy balance equation of the sum of the atmosphere column and the ocean mixed layer. In 2086-2100, the deep ocean does not reach a full equilibrium with the anthropogenic forcing but we have considered the effect of ocean heat flux using the term ΔO . Given the 15-years span of 2086-2100, the tendencies in the energy storage are very small and thus not included in the equations. For example, for a 20m ocean mixed layer, if it warms by 0.2 K over the 15-years span of 2086-2100, the tendency in the energy storage is only $C \cdot dT/dt = 20 \text{ m} \cdot 4186 \text{ J/kg} \cdot 1000 \text{ Kg/m}^3 \cdot 0.2 \text{ K} / (15 \cdot 365 \cdot 86400 \text{ s}) = 0.036 \text{ W/m}^2$.

Response to Reviewer #3:

Arctic amplification is an aspect of climate change that has widespread interest and has been the subject of many lines of inquiry concerning its mechanisms. This manuscript offers a new theoretical framework to the factors governing Arctic amplification that is novel and grounded in a fundamental aspect of how the atmosphere transports heat to the pole. It's a very illuminating new approach and I recommend publication upon a minor revision.

The manuscript deploys a well justified approach to atmospheric heat transport (AHT) to Arctic Amplification (AA). AHT's changes with climate can be thought of as depending on two mean-state quantities: the global-mean surface temperature, which increases the heat transport, and the temperature contrast in latitude, AA decreases this which is an offsetting factor. This is combined with imposed regional feedbacks and forcing to capture the behavior of comprehensive climate models.

Thank you for your insightful review. We have provided point-by-point responses to your comments below.

Two overarching points that should be acknowledged in the manuscript:

A. The lapse rate feedback is prescribed from the results of GCMs: the Arctic lapse rate is affected by a combination of factors, including radiative forcing and atmospheric heat transport. The theoretical understanding of this is clear about why different forcing and feedback have different lapse rate changes (e.g., Cronin and Jansen 2016), and there are feedback analysis approaches that are designed to separate this (e.g., Feldl et al 2020). So, in principle, the Arctic box's feedback depends on how its solution evolves. The empirical success of the authors' approach suggests this isn't a leading-order effect---good results despite the omission of this complicating factor. But it's important to communicate this aspect of the new framework introduced.

Following your suggestion, we have mentioned the intricacy in the Arctic lapse rate feedback (L199-201). Given the strong coupling in the changes of atmospheric temperature and humidity, the water vapor feedback would present similar intricacy. Our analyses focus on the compensation between the lapse-rate and water-vapor feedback, and we show that the combined feedback is nearly identical between the global mean and the Arctic mean.

B. The need to impose feedbacks from climate model simulation means this framework in

practice is still heavily diagnostic (vs. purely diagnostic approaches described near L96). This means the new framework is not predictive, despite the title and abstract's language.

Following your comment, we have removed the word “predictive” from the title and the article body. The reason we use “predictive” initially is that we have derived a mathematical formula of AA and it helps us “predictively” understand how AA changes quantitatively with parameters/feedbacks. We think this is beyond the general diagnostic framework but we agree that it is not a forecast model that can directly predict the future degree of AA.

Some clarifications:

1. The theory's "baseline AA" (L178) has a similar dependence on the global feedback parameter as moist energy balance model theory (Merlis and Henry 2018)---it appears in the denominator. Are these the same? Is the value of 1.67 (L181) the same?

Indeed, Merlis and Henry 2018 investigated polar amplification in an EBM with uniform forcing and feedback. The constant climate feedback they used (-B) is $-1.8 \text{ W m}^{-2} \text{ K}^{-1}$, which is consistent with $\lambda^G = -1.7 \pm 0.6 \text{ W m}^{-2} \text{ K}^{-1}$ here. The degree of the baseline AA (~ 1.67) is consistent with their EBM result (see their Fig. 1). We have mentioned this consistency in our manuscript (L184-186). Thank you for raising this point.

2. Fig. 2e shows the baseline AA is the largest factor, larger than the feedback part (gray vs. yellow). Is it fair to conclude that the energy transport related part of AA is dominant? In other words, in the diagnostic approaches that are critiqued in the introduction (Pithan & Mauritsen 2014), the AHT term is not dominant but one can't make firm conclusions based on that approach. Can one now make a firm conclusion on that basic question?

Yes, we now see that the role of AHT is represented by its partial sensitivity to global uniform warming (a) and Arctic enhanced warming (b). Even with uniform forcing and feedbacks, the baseline AA exists as $1 + \frac{a}{b - \lambda^G}$, which is about 1.67 for MME (gray bar). The effect of differential feedbacks increases the degree of AA by about 1 (yellow bar). The skill of I^{AA} in predicting AA is improved from $r=0.71$ to $r=0.92$ by using the model-dependent a and b , highlighting the important role of AHT in determining the degree of AA in individual models. We have further emphasized the role of AHT in the manuscript (L219-236).

Minor presentation revisions:

- The diagram in Fig. 1 shows two regions: Arctic and everywhere else, but the equations suggest the feedback parameter for "everywhere else" is actually the global mean. Is there this double counting where the diagnosed Arctic feedback enters both λ_A and G ?

We use "globe" (following Eq. 1 and 2) as one can drop the term AHT for the global mean energy balance. Using "globe" makes our derivation simpler and naturally connects to the definition of AA (the ratio between the Arctic mean and the global mean). The Arctic feedback does enter the global mean feedback, but the Arctic (poleward of 65°N) only accounts for 4.6% of the global area. We have modified the diagram of Fig. 1 to avoid inconsistency.

- L124 typo before which- paragraph ending

Corrected

L210: this sounds like a rehash of the central discussion of Held & Shell 2012 who suggest a different feedback decomposition that assumes constant relative humidity as the reference response, but applied to the zonal-mean. But others have presented zonal-mean feedbacks using that different decomposition, so I don't think this is particularly new, and maybe not worth emphasizing strongly particularly given the ambiguity of what the Arctic lapse rate is controlled by. For example, Hahn et al. 2021 did the diagnostic energy budget approach with both feedback decompositions and found: "As a result, the relative contribution of the lapse-rate feedback to Arctic amplification is weakened in the fixed-RH framework, with stronger contributions from the albedo feedback and poleward moisture transport." Using all of Fig. 3 on this point is overkill to me.

In this study, we show that the sum of the lapse-rate and water-vapor feedback is nearly identical between the global mean and the Arctic mean. As a result, the combined feedback contributes little to the degree of AA. Held & Shell 2012 showed that the lapse-rate and water-vapor feedbacks compensate with each other and their sum can be understood from the fixed relative humidity response. But this does not necessarily mean that their sum should be identical between the global mean and the Arctic mean, which is what we emphasize here. The fixed-RH diagnose approach provides different decomposition to individual feedbacks and an alternative way to understand feedbacks. This would not affect our conclusion here, which is about the

fundamental compensation between the lapse-rate and water-vapor feedbacks in their contributions to AA. Since the lapse-rate feedback has been long appreciated as a major contributing factor for AA, we think it is important to emphasize our result.

Following your comment, we have described the intricacy in the lapse-rate feedback, acknowledged previous work and better explained the contribution of our work (L197-217).

- L272 typo with lambda superscript, repeated G

Corrected

- L293 highlighting individual models by this coding is not the most effective way to communicate, perhaps point to the right side of Fig. 4 columns instead

We now mention the model name too. We used the model number so the readers can refer to the scatterplot figures. The correspondence between model number and name is provided in Fig. 2.

- L376: does this ocean heat uptake parameter agree with previous published results like Geoffroy et al. 2013 and work by J. Gregory? I had a ~50% more negative number in mind

Geoffroy et al. 2013 used a two-layer ocean model and defined the heat exchange coefficient γ as $H = \gamma(T - T_o)$, where T_o is the deep ocean temperature that warms up very slowly. Here, for simplicity and to focus on AA, our two-box model does not consider a two-layer ocean model and we define the ocean feedback parameter as $\lambda_o = \frac{\Delta O}{\Delta T}$, which is the same (but with opposite sign) as the ocean heat exchange coefficient κ defined in Gregory and Forster (2008). ΔO in our manuscript is the same as $-H$.

As pointed out by Geoffroy et al. 2013, “These values (γ , MME at **0.7 W m⁻² K⁻¹**) are somewhat larger than the zero-layer EBM heat exchange coefficient κ values estimated by Gregory and Forster (2008). One could expect that the introduction of the deep-ocean temperature perturbation T_o reduces the contribution of the temperature difference term to the deep-ocean heat uptake $H = \gamma(T - T_o)$ formulation (**for a given H : $T - T_o < T$, so that $\gamma > \kappa$**).”

Another difference is that we consider the SSP245 experiment while Geoffroy et al. 2013 consider

the 4xCO₂ experiment. The exact value of κ depends on which period and scenario one analyzes, as the more the slow response is presented (a larger ΔT_0), the lower κ is.

We have mentioned that λ_0 is similar to κ in Gregory and Forster (2008) in the main text (L139) and mentioned the subtle difference to Geoffroy et al. 2013 in Method (L486-488).

- eqn. S9, S10: there's never a saturation specific humidity defined or relative humidity stated that it is assumed constant

Indeed, the assumption of approximate unchanged relative humidity is used here. We have explained more clearly what relation and assumption we have used in deriving these two equations.

- Fig. 4 caption uses FF, but that's not defined in the main text
FF is defined in L256.

REVIEWERS' COMMENTS

Reviewer #1 (Remarks to the Author):

The authors have fundamentally addressed my comments from the first review, but I have a few remaining points that should be addressed before publication:

- 1) I am unconvinced of the description of AHT as somehow between forcing and feedback. While it is not a classic radiative feedback, it clearly is a response of the climate system to the external forcing mediated by and scaling with surface temperature change. It should thus be understood as a feedback, not as part of the forcing.
- 2) The derivation of a and b from model output lacks detail and thus clarity. What T and q are used? I would guess t_{as} and h_{uss} , i.e. 2m near-surface values, but this is not spelled out. If my guess is correct, are results similar when using 850 hPa values instead?

Lines 33-35: As there is some ambiguity in attributing the spread of AA to the different factors for individual models, this claim seems a bit overstated.

Line 50: given the references to Paeleoclimate, climate change does not have to be warming
l. 106 ff: The argument makes physical sense, but feedback analysis sometimes use global-mean temperature change even for regional feedback factors, in which case the argument would not hold. This should be discussed to avoid confusion.

Reviewer #2 (Remarks to the Author):

The article is worth publishing, it is an interesting study offering (yet another) tool for diagnosing Arctic Amplification (AA) in models. My main problem with the article is that the overall novelty of the article needs to be explained better. I am happy to see that they removed the word "predictive" from the article, but still, this is just a diagnostic framework for AA, and it is not even close a "theory". I would strongly suggest not using big words for describing this analysis. I am glad the authors included the sentence explaining why AHT will increase in the warmer climate (higher sensitivity of dq^*/dT in the tropics). This very important point was completely missing (I believe) in the first version. This has been employed before for explaining AA (e.g. Langen Alexeev, Climate Dynamics 2007). Having said that, still, there is no formula that will explicitly include any physics (e.g. Clausius-Clapeyron equation) to quantify AA dependence on sensitivity of AHT to changes in the temperature. I notice other reviewers point that some other parts of the paper just reformulate previous analyses, which is fine, but then what is new and exciting here? But again, the article represents a nice and interesting diagnostic study of AA in

models, which is why I think it is worth publishing. I'd give it a solid "B".

Reviewer #3 (Remarks to the Author):

Overall, the revised manuscript addresses the concerns I had with the presentation (e.g., is it a predictive or diagnostic framework?) and cleaned up the typos I found. I recommend accept with an optional minor revision.

The one outstanding point of clarification concerns the motivation for and discussion of Fig. 4b: the vertical axis is the difference between the atmospheric heat transport parameters 'a' and 'b'. The discussion of the theoretical formula is clear that Arctic Amplification increases with increasing 'a' and decreases with decreasing 'b', but is there a proper mathematical motivation for the difference 'a-b' as being the unique quantity to characterize intermodel spread? This is worthy of a clear discussion because 'a' and 'b' are approximately the same magnitude (the authors added more on theoretical derivations in the revised methods). Put another way, I don't see why the boundaries between the shaded colors in Fig. 4b are straight lines given the equation has $a/(b-\lambda_G)$. Shouldn't there be some curvature from the inverse, 'b' appearing in the denominator?

Response to Reviewer #1:

The authors have fundamentally addressed my comments from the first review, but I have a few remaining points that should be addressed before publication:

Thank you for your insightful comments. We have provided point-by-point response below and revised the manuscript accordingly.

1) I am unconvinced of the description of AHT as somehow between forcing and feedback. While it is not a classic radiative feedback, it clearly is a response of the climate system to the external forcing mediated by and scaling with surface temperature change. It should thus be understood as a feedback, not as part of the forcing.

According to the formula $\Delta AHT \cong a\Delta T^G - b(\Delta T^A - \Delta T^G)$, the term $b(\Delta T^A - \Delta T^G)$ represents the reduced AHT due to AA and is a negative feedback to AA. On the other hand, the term $a\Delta T^G$ represents the increase in AHT due to enhanced meridional moisture gradient from global uniform warming as dq^*/dT is higher in warmer tropics. $a\Delta T^G$ is independent of AA and works to warm the Arctic. So, we think $a\Delta T^G$ can be interpreted as a forcing to AA.

As you pointed out, this forcing part of ΔAHT is not a direct radiative forcing F . We have edited the related sentences (L117-121) as follows to avoid potential confusion.

The change in AHT, ΔAHT , is a unique factor involved in AA. It is neither a pure forcing (e.g., F) nor a pure feedback (e.g., λ_i) **to AA**. Instead, it consists of two parts -- a **forcing-like** part as global-scale warming enhances meridional moisture gradient (dq^*/dT is higher in warmer tropics) and increases AHT to amplify Arctic warming, and a negative feedback part as AA weakens meridional temperature gradient and reduces AHT.

2) The derivation of a and b from model output lacks detail and thus clarity. What T and q are used? I would guess tas and huss, i.e. 2m near-surface values, but this is not spelled out. If my guess is correct, are results similar when using 850 hPa values instead?

Yes, T and q here use 2m near-surface value. We have clarified this point in the text and in the figure caption. If one uses 850hPa T and q to derive a and b , one should also use 850hPa T for the AHT formula and the energy-balance equations (Eqs. 1,2,5), and define AA and diagnose feedback parameters according to the 850hPa T. While this is doable and one can develop an analytic theory of AA with respect to the 850hPa temperature, it may not be preferred compared to that of surface temperature.

Lines 33-35: As there is some ambiguity in attributing the spread of AA to the different factors for individual models, this claim seems a bit overstated.

In Fig. 4, we have attributed the intermodel variation of AA to the intermodel variation of the physical factors. First, we have decomposed the degree of AA into the baseline AA (I_B^{AA}) and the effect of differential forcing and feedback between the Arctic and the globe (FF) (Fig. 4a). Then, we show that the variation of I_B^{AA} is well explained by the variation in λ^G and d (Fig. 4b) and the variation in FF can be well explained by $\lambda^A - \lambda^G$ and λ^G (Fig. 4c). Finally, the intermodel variation in λ is further attributed to contributions of individual physical feedbacks (Fig. 4d).

We have replotted Fig. 4b using d instead of $a - b$. Unlike a and b , d can be accurately diagnosed for each model without ambiguity and a higher d implies a larger a and/or a smaller b . We have rewritten the text related to Fig. 2b to make this attribution clearer.

Considering we have assumed a constant parameter $\gamma \equiv \frac{F^A}{F^G}$, we have edited the sentence to “The formula captures the varying AA in climate models and attributes the spread to models’ feedback parameters and AHT physics”.

Line 50: given the references to Paleoclimate, climate change does not have to be warming

Following your comment, we have edited the sentence as “ Robustly seen in paleo proxy records^{1,2}, historical observations³⁻⁵, and model simulations⁶⁻⁹, the temperature response to climate change is amplified in the Arctic relative to the rest of Earth.”

I. 106 ff: The argument makes physical sense, but feedback analysis sometimes use global-mean temperature change even for regional feedback factors, in which case the argument would not hold. This should be discussed to avoid confusion.

Following your comment, we have mentioned in both the text (L94) and figure legend (Fig. 2b) that the feedback parameters are defined with respect to the Arctic and global mean warming, respectively.

The conventional diagnostic framework partitions the absolute warming (e.g., ΔT^A) to contributions of individual factors according to the following energetic equations,

$$F^A + \Delta O^A + \Delta AHT + \sum_i \lambda_i^A \Delta T^A = 0. \quad (R1)$$

By writing $\lambda_p^A = \lambda_p^G + \lambda'_p$, we have

$$F^A + \Delta O^A + \Delta AHT + (\lambda_{alb}^A + \dots + \lambda'_p) \Delta T = -\lambda_p^G \Delta T \quad (R2)$$

Then, by dividing $-\lambda_p^G$ on both sides, we have

$$\Delta T^A = \frac{F^A}{-\lambda_p^G} + \frac{\Delta O^A}{-\lambda_p^G} + \frac{\Delta AHT}{-\lambda_p^G} + \frac{\sum_i \lambda_i^A \Delta T^A}{-\lambda_p^G} \quad (R3)$$

In above derivation, λ_i^A must be defined with respect to the Arctic warming (multiplied by ΔT^A in the last term of RHS), otherwise one cannot move ΔT^A to the LHS and derive Eq. R3 which partitions the total warming ΔT^A to individual factors. Even if one ignores the derivation process and **define λ_i^A with respect to global mean warming**, the diagnosed contributions of a feedback (i.e., $\frac{\lambda \Delta T}{-\lambda_p^G}$) should not be changed, as $\lambda \Delta T$ is the radiative anomaly associated with that feedback and it does not depend on whether the feedback parameter λ is define with respect to regional or global mean warming. We have included Eq. R3 in the main text.

Response to Reviewer #2:

The article is worth publishing, it is an interesting study offering (yet another) tool for diagnosing Arctic Amplification (AA) in models. My main problem with the article is that the overall novelty of the article needs to be explained better. I am happy to see that they removed the word "predictive" from the article, but still, this is just a diagnostic framework for AA, and it is not even close a "theory". I would strongly suggest not using big words for describing this analysis. I am glad the authors included the sentence explaining why AHT will increase in the warmer climate (higher sensitivity of dq^*/dT in the tropics). This very important point was completely missing (I believe) in the first version. This has been employed before for explaining AA (e.g. Langen Alexeev, Climate Dynamics 2007). Having said that, still, there is no formula that will explicitly include any physics (e.g. Clausius-Clapeyron equation) to quantify AA dependence on sensitivity of AHT to changes in the temperature. I notice other reviewers point that some other parts of the paper just reformulate previous analyses, which is fine, but then what is new and exciting here? But again, the article represents a nice and interesting diagnostic study of AA in models, which is why I think it is worth publishing. I'd give it a solid "B".

Thank you for your insightful comments. The major contribution of this work is to establish **an analytic formula** for the degree of AA, which we believe **works beyond a diagnostic framework**. The formula allows us to predictively understand how AA will change if certain physical factors change, while a diagnostic framework cannot do that. Furthermore, the formula conveys a clear picture of how the degree of AA is mutually determined by multiple factors and articulates the intricate role of AHT. Such understanding cannot be gained from the existing diagnostic framework (see our discussion in L99-113). Besides that, the formula attributes the intermodel spread of AA to physical factors, which can be further investigated by modelling centers. Also, we have now added a paragraph to show that the formula can analytically explain the results of previous numerical studies which artificially manipulate the feedbacks and forcing of AA (L310-327).

We have explained these contributions throughout the paper and highlighted what the new insights are in the Summary and Discussion sections. We have added the reference

of Langen Alexeev (2007) for the point that global-scale warming would increase AHT. But please note that AHT is a combination of forcing and feedback to AA and it can either increase or decrease in a warmer climate depending on the model.

Response to Reviewer #3:

Overall, the revised manuscript addresses the concerns I had with the presentation (e.g., is it a predictive or diagnostic framework?) and cleaned up the typos I found. I recommend accept with an optional minor revision.

Thank you for your insightful comments. We have provided point-by-point response below and revised the manuscript accordingly.

The one outstanding point of clarification concerns the motivation for and discussion of Fig. 4b: the vertical axis is the difference between the atmospheric heat transport parameters 'a' and 'b'. The discussion of the theoretical formula is clear that Arctic Amplification increases with increasing 'a' and decreases with decreasing 'b', but is there a proper mathematical motivation for the difference 'a-b' as being the unique quantity to characterize intermodel spread? This is worthy of a clear discussion because 'a' and 'b' are approximately the same magnitude (the authors added more on theoretical derivations in the revised methods).

While a/b should not be too far from 1, $a-b$ still varies considerably among models, and its value can be either negative or positive. $a - b$ is linked to $d = \frac{\Delta AHT}{\Delta T^G} - \frac{\widehat{\Delta AHT}}{\Delta T^G}$ (i.e., the distance to the regression line in Fig. 2a) but may not be further simplified. A higher d implies a larger a and/or a smaller b , that is, a higher $a - b$.

Put another way, I don't see why the boundaries between the shaded colors in Fig. 4b are straight lines given the equation has $a/(b-\lambda_G)$. Shouldn't there be some curvature from the inverse, 'b' appearing in the denominator?

Indeed, the exact relation between $\frac{a}{b-\lambda_G}$ and $a - b$ can be rather complicated with b in the denominator. In the previous version, we have used the following approximate formula to plot the shading in Fig. 4b,

$$\frac{a}{b - \lambda_G} = \frac{\hat{a} + x/2}{\hat{b} - x/2 - \lambda_G}$$

where $x = (a - b) - (\hat{a} - \hat{b})$.

Now, inspired by your comment, we have related the baseline AA $1 + \frac{a}{b - \lambda_G}$ to $d = \frac{\Delta AHT}{\Delta TG} - \frac{\widehat{\Delta AHT}}{\Delta TG}$ (instead of $a - b$) and λ_G . The benefit is that, unlike $a - b$, d can be accurately diagnosed for individual models. Please see Fig. 4b and the related paragraph (L268-276) for more details. The shading shows $\frac{\hat{a} + d}{\hat{b} - \lambda_G}$ and helps visualize the dependency of the baseline AA on d and λ_G . We have explicitly stated how the shading is calculated in the figure caption.